# Assessing glacier melt contribution to streamflow at Universidad Glacier, central Andes of Chile

Claudio Bravo[1], Thomas Loriaux[1,2], Andrés Rivera[1,3], Ben W. Brock[4]

[1] Centro de Estudios Científicos, Valdivia, Chile.
[2] School of Earth Sciences, University of Bristol, Bristol, UK.
[3] Departamento de Geografía Universidad de Chile, Santiago, Chile.
[4] Department of Geography, Northumbria University, Newcastle, UK.

*Correspondence to:* Claudio Bravo (claudiobravo.lechuga@gmail.com)

**Abstract**. Glacier melt is an important source of water for high Andean rivers in central Chile, especially in dry years when it can be an important contributor to flows during late summer and autumn. However, few studies have quantified glacier melt contribution to streamflow. To address this shortcoming, we present an analysis of meteorological conditions and ablation for Universidad Glacier, a large valley glacier in the central Andes of Chile at the head of the Tinguiririca River, for the 2009-2010 ablation season. We used meteorological measurements from two automatic weather stations installed on the glacier to drive a distributed temperature-index and runoff routing model. Total modelled glacier melt is compared with river flow measurements at three sites located between 0.5 and 50 km downstream. The temperature-index model was calibrated at the lower weather station site and showed good agreement with melt estimates from an ablation stake and sonic ranger, and with a physically-based energy balance model. Universidad Glacier is characterized by extremely high melt rates over the ablation season which may exceed 10 m water equivalent on the lower ablation area, representing between 10% and 13% of the mean monthly streamflow at the outlet of the Tinguiririca River Basin between December 2009 and March 2010. This contribution rises to a monthly maximum of almost 20% in March 2010 demonstrating the importance of glacier runoff to streamflow, particularly in dry summers such as 2009-2010. The temperature-index approach benefits from the availability of on-glacier meteorological data, enabling the calculation of the local hourly-variable lapse rate, and is suited to high melt regimes, but would not be easily applicable to glaciers further north in Chile where sublimation is more significant.

## 1 Introduction

The central region of Chile (30° - 37° S), in southern South America, is characterized by a high dependence on the water supply coming from the Andes. This region, incorporating the capital city, Santiago, has more than 10 million inhabitants representing 60% of the country's population. In addition to domestic supply, water is a crucial resource for agriculture irrigation, industries, mining, hydropower generation, tourism and transport (Aitken et al., 2016; Masiokas et al., 2006; Meza et al., 2013; Ayala et al., 2016, Valdés-Pineda et al., 2014). Population growth and urban expansion in recent years are increasing the demographic pressure on water resources (Meza et al., 2012).

In this region, winter precipitation is driven by the interactions between the westerlies circulation and the Andean natural barrier, and summer runoff is strongly influenced by the storage and release from glaciers and snow covers (Garreaud, 2013). Accurate knowledge of the processes involved in the runoff generation from mountainous areas is vital to understand and predict the availability of water resources and contribution to sea level rise (Mernild et al., 2016) especially considering the ongoing and projected future decrease in glacier volume under climate warming scenarios (Pellicciotti et al., 2014, Ragettli et al., 2016).

In these latitudes, the Andes present several peaks over 6000 m above sea level (asl) and have a mean elevation of ~4000 m asl . The majority of annual precipitation occurs during the winter months, which accumulates as snow above the winter 0°C isotherm altitude, between 1500 and 3500 m asl (Garreaud, 2013). This seasonal snowpack provides an important water reservoir for the following summer months, when warm temperatures and high incoming solar energy cause the melting of snow. As a consequence, rivers in the high Andes basins of central Chile are mainly driven by the melting of the seasonal snowpack (Cortés et al., 2011). However, another key source of water in the summer dry season is the presence of glaciers along the Andes Cordillera. Crucially, glacier melt is an important source of water for Andean rivers in dry summers when little or no precipitation occurs at the upper watersheds and the seasonal snowpack is exhausted (Gascoin et al., 2010, Masiokas et al., 2013, Ohlanders et al., 2013). For example, Peña and Nazarala (1987) estimated that the contribution of ice melt to the high basin of the Maipo River (5000 km$^2$, outlet at 850 m asl) in the 1981/1982 summer was highest in February and represented 34% of total streamflow.

There have been only a few physically-based distributed glacio-hydrological modelling investigations in the Andes of Chile (Pellicciotti et al., 2014; Ayala et al., 2016), which is an important limitation for the understanding of future glacier contribution to river flows, considering the current trends of glacier shrinkage (e.g. Bown et al., 2008; Le Quesne et al., 2009; Malmros et al., 2016) and negative mass balance (Mernild et al., 2015) in the region. One of the most studied glacier in the region is Juncal Norte Glacier, where Pellicciotti et al. (2008) investigated the point scale energy balance and melt regime using an automatic weather station (AWS) located on the glacier ablation zone, showing that the ablation process is dominated by incoming shortwave radiation. Using a physically-based distributed glacier-hydrological model, Ragettli and Pellicciotti (2012) estimated that melted glacier ice from Juncal Norte Glacier contributed 14% of the basin (241 km$^2$, 14% glacierized, outlet at ~2250 m asl) streamflow for the entire hydrological year 2005/2006, with a maximum of 47% over the late ablation season (February to April). Despite these advances, such results are limited to one basin and cannot necessarily be extrapolated, particularly along climatic gradients to the north and south. Other glacier energy balance studies in central Chile have focused on improving understanding of energy fluxes and ablation at the point scale (Corripio and Purves, 2004; MacDonnell et al., 2013) or on the impact of volcanic ash on energy balance and melt (Brock et al., 2007; Rivera et al., 2008). There is therefore a lack of knowledge of spatial and temporal melt patterns at the glacier-wide scale and of glacier melt contribution to downstream discharge over a full ablation season.

We present an analysis of meteorological conditions and ablation for Universidad Glacier, a large valley glacier in central Chile, located in a climatic transition zone with a Mediterranean climate type, between the humid temperate south and arid north of the country. The main aims are: (1) to identify the principal meteorological drivers of ablation and their patterns and trends during a full ablation season; (2) to compare methods of ablation estimation using two models of differing complexity and input data requirements; and (3) to estimate the contribution of glacier melt to downstream river flows and its water resource implications. The aims are addressed using point energy balance and distributed temperature-index models forced with data from two AWS located on the glacier ablation and accumulation zones, and stream gauging records both proximal to the glacier snout and 50 km downstream at mid-altitude on the Tinguiririca River.

## 2 Data and methods

### 2.1 Study area

Universidad Glacier (34° 40' S, 70°20' W) is located in Central Chile, in the upper part of the Tinguiririca Basin (1436 km$^2$), 55 km east of San Fernando city and 120 km south-east of Santiago (see Fig. 1 for location). The upper Tinguiririca Basin is defined as a snowmelt dominated river (Cortez et al., 2011) with runoff peak occurring between November and January (Valdés-Pineda et al., 2014). The area of the glacier is 29.2 km$^2$ with a length of 10.6 km and an altitudinal range of 2463 m asl to 4543 m asl (Le

Quesne et al., 2009). The glacier has an accumulation zone divided into two basins which converge at an altitude of ~2900 m asl. Below this elevation, the glacier has a well-defined tongue. The equilibrium line altitude (ELA) for the 2009-2010 hydrological year, based on the position of the end of summer snowline was located in the range between 3500 and 3700 m asl depending on the aspect of the glacier (Fig. 1). The general aspect is southerly, but the west accumulation zone has an easterly aspect. Universidad Glacier is a valley glacier that is part of a more extensive glacier complex, which includes the Cipreses Glacier flowing to the north, Palomo Glacier flowing to the north-east, Cortaderal Glacier flowing to the east, and other small glaciers flowing to the west. Another feature of the basin is the presence of small lakes mainly associated with glaciers (proglacial lakes) and debris-covered glaciers (supraglacial lakes).

Scientific investigations at Universidad Glacier were initiated by Lliboutry (1958) who described some morphological characteristics of the glacier surface including ogives, blue bands, penitents and moraines, noting the absence of penitentes above 3800 m asl, in contrast to glaciers further north in Chile. According to his observations, the lower part of the glacier had a sudden advance around 1943. After this event, a spectacular recession from 1946-1950 was recorded. More recently, a frontal retreat of 1000 m for the period 1955-2007 was documented from aerial photographs, historical documents, tree ring chronologies and satellite images (Le Quesne et al., 2009). Wilson et al. (2016) estimated Universidad Glacier surface velocities between 1967 and 2015 and identified an increase in surface velocities between 1967 and 1987, followed by a deceleration between 1987 and 2015. Furthermore, a cumulative frontal retreat of $465 \pm 44$ m was found between 1967 and 2015.

## 2.2 Experimental setting

The study focuses on the ablation season (1 October to 31 March) of the 2009/2010 hydrological year, when the discharge, meteorological and glaciological conditions were monitored. We focused on one ablation season due to availability of data. The 2009/2010 hydrological year is of significance as it marks the beginning of a period of extreme aridity (2010-2015) in central and southern Chile (Bosier et al., 2016) which extended into 2017 according to data from the Dirección General de Aguas de Chile (DGA) and Dirección Meteorológica de Chile.

Data collected include meteorological observations at two AWS, surface lowering monitoring from ablation stakes and a sonic ranger (Fig. 1), satellite-derived snow cover distribution and discharge measurements in the proglacial stream. Following the analysis of energy fluxes at the location of the lower AWS, a temperature-index model was calibrated and applied at the glacier scale. The resulting melt amounts were used to estimate total glacier discharge, which is compared with downstream discharge records.

## 2. 3 Automatic weather stations (AWS)

Two AWS were installed on the surface of the glacier (Fig. 1). One on the ablation zone (AWS1, 34° 42' S, 70° 20' W, 2650 m asl) and the second one on the accumulation zone (AWS2, 34° 38′S, 70°19' W, 3626 m asl). AWS1 recorded a full set of energy balance variables including air temperature, humidity, wind speed and direction, net all-wave radiation, incoming shortwave radiation and atmospheric pressure, while AWS2 recorded the same variables but omitted radiation measurements. Although AWS1 was installed at the beginning of 2009 we restricted the analysis to the ablation season defined as 1 October 2009 to 31 March 2010. AWS2 recorded data from 10 December 2009 to 31 March 2010. Both AWS recorded data averaged at a 15-minute interval; however, we use hourly mean values as model inputs.

## 2.4 Ablation measurements: stakes and sonic ranger

Three stakes installed on the ablation zone of the glacier between 30 September and 3 October 2009 were read on 21 November while the surface was still snow covered at each stake (Fig. 1, Table 1). Stake 1 was located close to AWS1 and was used to assess point melt estimations from the different models. Snow density was measured using the standard Mount Rose procedure (U.S. Department of Agriculture, 1959) on the days of installation and re-measurement of stakes. We calculated the mean snow density (Table 1) and water equivalent (w.e.) surface ablation for each stake.

A Campbell Scientific SR-50 sonic ranging sensor was installed next to AWS1. The sensor recorded surface lowering continuously every 15 minutes during a 73 day period. SR-50 data were filtered using a Hampel filter (Pearson, 2002) and then hourly means were calculated. Lowering measurements were converted to w.e. ablation values using snow density measured at stakes (Table 1).

## 2.5 Snowline elevation estimation using MODIS snow product

To derive snowline elevation, we used the MODIS/Terra L3 global daily snow cover product (MOD10A1, Hall et al., 2002) with a spatial resolution of 500 m, which retrieves subpixel fractional snow cover area. MOD10A1 was developed using a regression with Landsat TM (30 m spatial resolution) Normalized Difference Snow Index (NDSI), offering a much more accurate approach for detecting snow covered area than previous satellite snow-cover products (Cortés et al., 2014). In order to map the snow line throughout the monitored period, we obtained the hypsometric curve of Tinguiririca Basin from an ASTER GDEM V2 with a resolution of 30 m (Tachikawa et al., 2011) and then calculated the snowline altitude for the austral summer of 2009-2010 in the upper Tinguiririca Basin. The MODIS snow cover product was used only if the cloud fraction for each satellite image was less than 30%. The snowline elevation on days of high cloud cover was estimated using a linear interpolation between the last day before and the first day after the data gap. The time series of snowline elevation is used as a model input to define snow or ice surface areas on the glacier. We used the MOD10A1 product since it provides a reliable differentiation of the ice surface of Universidad Glacier, which is partially covered by debris and aerosols. However, the MOD10A1 product gives the fractional snow cover for each pixel in the range 0 to 100, and to assure a correct snowline altitude we assumed the presence of snow in the pixel only when the fractional value was 100. However we acknowledge some uncertainty in the snowline altitude.

## 2.6 Degree-hour model (DHM)

We applied a standard degree-day model (DDM) (e.g. Hock, 2003, 2005) at an hourly time step, in order to estimate glacier surface melt during the 2009/2010 ablation season. The model was forced with hourly temperature data from AWS1.

Melt is calculated by multiplying the hourly positive temperature, $T_h^+$ by a factor that relates temperature and melt, referred to as the degree-day factor ($F_{DD}$), or degree-hour factor ($F_{DH}$) when applied at an hourly interval (De Michele et al., 2013). We used the stake 1 ablation measurement (Table 1) and the mean positive air temperature (4.6 °C) at AWS1 to estimate a $F_{DH}$ for snow. The percentage of hours with positive temperatures was close to 75%, therefore we only used time steps with positive values. Dividing the ablation value by the mean positive air temperature (Braithwaite et al., 1998), we obtained a $F_{DH}$ for snow of 0.12 mm w.e. °C$^{-1}$ h$^{-1}$. The $F_{DH}$ is multiplied by the positive hourly temperature at an hourly interval and results are summed up for every day. We did not have ablation stake measurements in the period when the ice surface was exposed, so we instead calibrated the $F_{DD}$ for ice based on melt estimated from the sonic ranger for the period after the 21 November, when field observations confirmed the site was snow free, to 10 December, the end of the sonic ranger record. The resulting $F_{DD}$ value was close to 8 mm w.e. °C$^{-1}$ d$^{-1}$, but to account for uncertainty due to the short period of ablation data on ice we applied a range of $F_{DD}$ values between 7 mm w.e. °C$^{-1}$ d$^{-1}$ and 9 mm w.e. °C$^{-1}$ d$^{-1}$, which corresponds to the mid-range of values for glacier ice reported in the review of Hock (2003).

The hourly $F_{DH}$ for ice was calculated by dividing the ice $F_{DD}$ by 24 to give $F_{DH}$ values for ice of 0.29 mm w.e. °C$^{-1}$ h$^{-1}$ and 0.38 mm w.e. °C$^{-1}$ h$^{-1}$, respectively.

Melt, $M$ (mm w.e. h$^{-1}$), is estimated by the following relationship:

$$M(t,z) = F_{DH}T_h^+(t,z),$$  (1)

The $F_{DD}$ values for ice were calibrated for daily average temperature and therefore could lead to a melt overestimation when applied as $F_{DH}$ values in the hourly model, since calculations are only made for hours with positive temperature. To test this potential bias, we compared melt calculations between a standard degree-day model and the DHM, using a $F_{DD}$ of 9 mm w.e. °C$^{-1}$ d$^{-1}$ and $F_{DH}$ of 0.38 mm °C$^{-1}$ h$^{-1}$, for the period of ice exposure at AWS1 and AWS2, representing the ablation and accumulation zones, respectively. At AWS1, the difference between daily and hourly model results is negligible (<2 mm w.e. out of a total of >8000 mm w.e.). This small difference reflects the almost continuously positive air temperature in the lower ablation zone during the study period (Fig. 2). At AWS2, the DHM overestimation is more significant at 290 mm w.e. out of a total of ~2000 mm w.e. of ice melt, representing an increase in melt of 15% over the DDM. This is due to more frequent negative temperatures in the accumulation zone during summer months. Melt overestimation in the accumulation zone will have a relatively small impact on total glacier runoff, which is dominated by melt from the ablation zone. Furthermore, there will be no melt estimation bias on snow as the $F_{DH}$ for snow was calibrated using field measurements from Universidad Glacier. We therefore apply the DHM in our calculations as it has the advantage of enabling hourly variations in temperature lapse rate to be accounted for in the distributed melt calculations across the glacier (next section).

## 2.7 Distributed degree-hour model (DDHM)

To distribute the DHM (distributed DHM, DDHM hereafter) we calculated the temperature lapse rate (LR) using both AWS in the common period (Fig. 2). Following the recommendation of Petersen and Pellicciotti (2011), we estimated a daily LR cycle (Fig. 3) considering that melt occurs mostly during the day. The mean hourly LR on an average day, oscillates between -0.004 °C m$^{-1}$ and -0.007 °C m$^{-1}$. During the night (24:00 h to 08:00 h local time) the mean temperature gradient was close to -0.006 °C m$^{-1}$ and fairly constant. During the day the LR has two cycles with minima in magnitude close to -0.005 °C m$^{-1}$ at 11:00 h and -0.004°C m$^{-1}$ at 19:00 h, separated by a maximum of -0.007 °C m$^{-1}$ at 16:00 h. While the LR minima are likely to be related to the strengthening of katabatic flow during the daytime (Petersen and Pellicciotti, 2011), the afternoon maximum is potentially caused by the erosion of the katabatic boundary layer on the lower glacier tongue, due to warm air advection from bare rock surfaces at the glacier sides and proglacial area (van de Broeke, 1997; Ayala et al., 2015).

Using the hourly LR, we distribute air temperatures over the entire glacier surface on a 30 m grid at an hourly time step, using the ASTER GDEM V2 and the glacier outline which was digitized from an ASTER image of 27 March, 2010. For October and November we assumed the same hourly lapse rate observed in the common period (December to March). Calculated melt values were not adjusted for reduction under debris cover on a medial moraine in the ablation zone.

## 2.8 Energy balance model (EBM)

A point scale energy balance model (EBM hereafter) was applied using weather station data collected at the AWS1, between 1 October 2009 and 29 January 2010. We restrict use of data only up until this date because a sharp change in net radiation and incoming shortwave radiation occurred after 29 January; therefore data from late January onwards are of questionable accuracy. Energy available for ablation, $\psi$ (W m$^{-2}$) was determined following Oerlemans (2010):

$$\psi = S_{in} + S_{ref} + L_{in} + L_{out} + H_s + H_l,$$  (2)

Where $S_{in}$ and $S_{ref}$ are incoming and reflected solar shortwave radiation, $L_{in}$ and $L_{out}$ are incoming and outgoing longwave radiation and $H_s$ and $H_l$ are the turbulent fluxes of sensible and latent heat, respectively. In this study, the conductive heat flux is considered negligible due to the predominantly positive air temperatures (Fig. 2) and, as summer precipitation totals amount are small, the amount of sensible heat brought to the surface by rain or snow is neglected (e.g. Oerlemans and Klok, 2002). The balance of the radiative fluxes $S_{in}$, $S_{ref}$, $L_{in}$ and $L_{out}$ was directly measured by the net radiometer sensor at AWS1. The sensible heat fluxes were calculated using the bulk approach (Cuffey and Paterson, 2010):

$$H_s = \rho_a c_a C^* u[T - T_s] (\Phi_m \Phi_h)^{-1}, \tag{3}$$

$u$ is wind speed in m s$^{-1}$, $T$ is air temperature in $K$ and $T_s$ ice surface temperature which is assumed to be a constant of 273.15 K (0°C). $C^*$ is a dimensionless transfer coefficient, which is a function of the surface aerodynamic roughness ($z_o$), assumed to be 0.001 m for melting snow and 0.01 m for ice on mid-latitude glaciers (Brock et al., 2006):

$$C^* = \frac{k^2}{ln^2 \left(\frac{z}{z_o}\right)}, \tag{4}$$

$z$ is the height above the surface of the $T$ and $u$ measurements (2 m) and $k$ is the von Kárman's constant (0.4). $\rho_a$ is the density of air which depends on atmospheric pressure $P$ (in Pa):

$$\rho_a = \rho_a^o \frac{P}{P_0}, \tag{5}$$

where $\rho_a^o$ (1.29 kg m$^{-3}$) is the density at standard pressure $P_0$ (101300 Pa). Finally, $c_a$ is the specific heat of air at a constant pressure (J kg$^{-1}$ K$^{-1}$) calculated as (Brock and Arnold, 2000):

$$c_a = 1004.67 \left(1 + 0.84 \left(0.622 \left(\frac{e}{P}\right)\right)\right), \tag{6}$$

The latent heat flux $H_l$ is:

$$H_l = \frac{0.622 \, \rho_a \, L_{v/s} C^* u[e - e_s]}{P} (\Phi_m \Phi_h)^{-1}, \tag{7}$$

where $e$ is air vapour pressure, $e_s$ is the vapour pressure at the glacier surface which is assumed to be 611 Pa (Brock & Arnold, 2000) the vapour pressure of a melting ice surface, and $L_{v/s}$ is the latent heat of vaporization or sublimation, depending on whether the surface temperature is at melting point (0°C) or below melting point (<0°C), respectively. Due to the absence of snow temperature measurements, the air temperature is assumed to determine the condition of evaporation or sublimation over the surface.

$e$ is obtained from the observed relative humidity at AWS1 ($f$) and using the empirical formula of Clausius-Clapeyron (Bolton, 1980), which is only function of air temperature ($T$ in °C):

$$e_{sat}(T) = 6.112 \exp\left(\frac{17.67 \, T}{T + 243.5}\right), \tag{8}$$

where $e_{sat}$ denotes the saturation vapour pressure in the air. Finally, $e$ is found by rearranging the following equation:

$$f = 100 \left(\frac{e}{e_{sat}}\right), \tag{9}$$

The melt rate ($M$) is calculated using:

$$M = \frac{\psi}{L_m \rho_w}, \tag{10}$$

Only positive values of $\psi$ are used in this equation. $L_m$ is the latent heat of fusion and $\rho_w$ is the water density (1000 kg m$^{-3}$). The sublimation rate ($S$) is calculated as (Cuffey and Paterson, 2010):

$$S = \frac{H_l}{L_s \rho_w}, \tag{11}$$

where $L_s$ is the latent heat of sublimation.

Stability corrections were applied to turbulent fluxes using the bulk Richardson number ($Ri_b$), which is used to describe the stability of the surface layer (Oke, 1987):

**for $Ri_b$ positive (stable):** $(\Phi_m \Phi_h)^{-1} = (\Phi_m \Phi_v)^{-1}$

$$= (1 - 5 Ri_b)^2, \tag{12}$$

**for $Ri_b$ negative (unstable):** $(\Phi_m \Phi_h)^{-1} = (\Phi_m \Phi_v)^{-1}$

$$= (1 - 16 Ri_b)^{0.75}.$$

$Ri_b$ is used to describe the stability of the surface layer:

$$Ri_b = \frac{g(T - T_s)(z - z_0)}{T u^2}, \tag{13}$$

where $g$ is the acceleration due to gravity.

**2.9 Proglacial discharge estimation**

The estimation of the river discharge was based on the determination of the cross section geometry and the monitoring of water level in the proglacial stream. Water level in the stream was monitored using a submersible pressure transducer (KPSI Series 500), installed 500 meters downstream of the glacier terminus (2428 m asl), which registered hourly water levels from 24 November 2009, until 14 April 2010. The proglacial stream receives the waters draining from a catchment with a total area of 86 km$^2$ which is partially covered by the Universidad Glacier (29.2 km$^2$) and some debris-covered ice bodies (4.4 km$^2$) (DGA, 2011).

In order to convert automatic water level measurements into discharge, we applied the widely used Manning's equation (Phillips and Tadayon, 2006; Fang et al., 2010; Gascoin et al., 2010; Finger et al., 2011) which combines environmental parameters such as stream slope, bed roughness and river section shape and area, for uniform open channel flow. It defines the discharge $Q$ [m$^3$ s$^{-1}$] as follows:

$$Q = VA, \tag{14}$$

where $A$ is the area of the cross section and $V$ is the average instantaneous velocity in the channel defined as:

$$V = \frac{1}{n} R^{\frac{2}{3}} \alpha^{\frac{1}{2}}, \tag{15}$$

where $R$ is the hydraulic radius, $\alpha$ is the slope of water surface, and $n$ is the Manning coefficient of roughness.

The geometry of the channel cross section was measured in the field at the location of the pressure transducer. The hydraulic radius is a measure of channel flow efficiency and is defined as the ratio of the cross sectional area to its wetted perimeter. We used the ASTER GDEM of 30 m resolution to estimate a slope of 0.03° for water surface in the gauged section. The roughness coefficient was set as 0.05, according to the United States Geological Survey (USGS) value for cobble and boulder bedrocks (Phillips and Tadayon, 2006), which corresponded to our site. The area of the cross section $A$ was estimated using water level observations from the pressure transducer and the width of the wet section, which in turn is estimated from an empirical relationship with water level.

We also make use of two other streamflow gauge measurements (see Fig. 1). The first is operated by a private company, Pacific HydroChile, located 1700 m from the glacier snout recording data every hour. The second one is operated by DGA, and is located on the Tinguiririca River at 560 m asl, 50 km downstream the Universidad Glacier. The contributing watershed to this lower gauge has an area of 1436 km$^2$ with a total ice cover of 81 km$^2$ (DGA, 2011), among which Universidad Glacier is by far the largest single ice body.

## 2.10 Discharge routing

At each grid cell and time step, glacier melt obtained with the DDHM was transformed into discharge using a linear reservoir model (Baker et al., 1982; Hock and Noetzli, 1997). For hourly time intervals, the proglacial discharge $Q$ is given by:

$$Q(t_2) = Q(t_1)e^{-1/K} + M(t_2) - M(t_2)e^{-1/K}, \tag{16}$$

where $M(t)$ is the rate of water inflow to the reservoir, which is considered to be equivalent to the total glacier melt. $K$ is the factor of proportionality in hours and is estimated from the time it takes for the water entering the top of the reservoir to flow out of the bottom (Baker et al., 1982). Using the record from the pressure transducer, the optimal value of $K$ was identified as 14.

## 3 Results

### 3.1 Meteorological and snow conditions

Time series of meteorological variables are shown in Fig. 2. During the December-March period, air temperature is almost constantly above 0°C at AWS1, but negative nocturnal values are more frequent at AWS2.

Wind speed shows some inter-daily variability, but hourly values are predominantly between 2 m s$^{-1}$ and 8 m s$^{-1}$. Wind speed was generally lower in summer (December to March) than spring (October to November). The prevailing wind direction (~10° to ~45°) corresponds to the general ice flow direction (Fig. 4), indicating a persistent katabatic wind. Wind direction in the accumulation zone (not shown) also shows a predominant katabatic flow aligned with the ice flow direction. The daily cycle of wind direction at AWS1 reveals that the prevailing katabatic wind is slightly weakened during afternoon hours (between 14:00 h to 18:00 h, local time, Fig. 4), corresponding with a temporary strengthening of the daytime temperature lapse rate (Fig. 3). Relative humidity shows a large daily variability (Fig. 2). Saturation is reached on several days in the period.

The snow line altitude derived from MODIS data is shown in Fig. 5. At the beginning of the ablation season, the entire glacier surface was covered by snow. The snowline altitude increased gradually until mid-January and thereafter stabilized between 3800 and 4000 m asl. There is some variability in the snow line position, probably due to varying proportions of cloud cover on different days. This snowline altitude range, derived from the MODIS MOD10A1 snow cover product (Section 2.5) is slightly higher than the altitude of the ELA estimated with the ASTER image from the end of March of 2010 (3500 to 3700 m asl, Fig. 1) possibly due to differences in spatial resolution in the two types of imagery. In the first half of the ablation season a high percentage of cloud cover (greater than 30%) affected snowline detection.

### 3.2 Energy balance

Figure 6 shows the daily mean of observed energy fluxes (net radiation and incoming shortwave radiation), turbulent fluxes calculated by the EBM (latent and sensible heat) and the resulting energy available for melt at AWS1, calculated by the model. Daily mean melt energy closely matches daily mean net radiation through much of the ablation season due to compensation between generally positive $H_s$ and mainly negative $H_l$, except during warm periods such as late January when $H_l$ turns positive

(Fig. 6, Table 2). Energy available for melt is highest in December and January when both incoming shortwave radiation (Table 2) and air temperature (Fig. 2) are high.

### 3.3 Point scale ablation comparison: observation and modelling

Sonic ranger measurements and stake observations (Fig. 1) were compared to melt estimated with the EBM and DHM at the location of AWS1 (Fig. 7). Sublimation represents a small percentage (2.8%) of the total ablation calculated with the EBM reflecting the predominantly positive air temperatures and, hence, a melt regime. Snow disappears at this location (~ 2650 m asl) around 21-22 November 2009.

Melt simulations from the DHM and EBM agreed well with the stake and sonic ranger ablation measurements. The DHM tended to lag behind the EBM and sonic ranger until 21 November, after which the EBM and sonic ranger estimates fall within the DHM range for $F_{DH}$ values between 0.29 mm w.e. h$^{-1}$ °C$^{-1}$ and 0.38 mm w.e. h$^{-1}$ °C$^{-1}$. The DHM estimated little or no melt during cold periods, e.g. the first 10 days of November, whereas the EBM indicates melt (as does the sonic ranger) caused by high insolation. During warm periods, e.g. 11-16 November, the DHM estimated higher melt rates than the sonic ranger sensor, indicating the high sensitivity of the DHM to temperature fluctuations. At the end of the comparison series, the EBM and sonic ranger total melt are within the range of the values estimated by the DHM. Overall, despite uncertainties in snow density and melt model parameters, the good agreement between the different models and measurements, supports the use of the DDHM to estimate total glacier melt.

### 3.4 Distributed degree hour model (DDHM)

Figure 8 shows the accumulated melt for each pixel of Universidad Glacier estimated by the DDHM during the period 1 October 2009 to 31 March 2010 using ice $F_{DH}$ values of 0.29 mm h$^{-1}$ °C$^{-1}$ and 0.38 mm h$^{-1}$ °C$^{-1}$. As the degree-hour melt is only a function of temperature, the higher zones of the glacier presented the lowest melt and *vice versa*. The maximum values of ~11000 mm w.e. (for $F_{DH}$ = 0.38 mm h$^{-1}$ °C$^{-1}$) were located on the lower glacier tongue (Fig. 9). All parts of the glacier experienced melting with totals around 1 m w.e. in the upper accumulation area. Bare ice surfaces accounted for ~85% of the total melt. As is expected, differences in the cumulative melt calculated with the two ice $F_{DH}$ values are higher in the tongue of the glacier (> 2000 mm w.e.) where ice is exposed for most of the ablation season.

### 3.5 Discharge

During the study period we estimated an average stream flow of 12 m³ s$^{-1}$ with a range from 4 m³ s$^{-1}$ and 43 m³ s$^{-1}$ (Fig. 10). Discharge values increased gradually between the end of November and the end of December. The mid ablation season (January and February) experienced two major discharge peaks. Subsequently, values decreased from late February to the end of March to values similar to those at the end of October (Fig. 10).

The hourly mean hydrographs have strong daily amplitude cycles during the high discharge months (Fig.10) and exhibit a characteristic shape for a glaciated catchment, with a steep rise and gradual decline (Nolin et al., 2010; Willis, 2011). Discharge peaked typically at 16:00 h, from a minimum at 10:00 h which, considering the large size of the glacier, indicates an efficiently channelized drainage system flow.

At the hourly scale, water discharge estimated at the HydroChile station showed high correlation with the values derived from the water pressure sensor installed near the glacier front (r=0.92). Generally, the HydroChile station values exceeded water discharges estimated from the water pressure sensor before mid-January; thereafter the water pressure sensor derived values exceeded the HydroChile results, until 27 February when there was a large earthquake in central Chile. A sudden jump in HydroChile values

occurred around this date, most likely due to this earthquake, whereas the pressure sensor derived values were adjusted for the change in water height. For comparison purposes we rejected data from the HydroChile station after the earthquake.

## 3.6 Comparison of glacier melt water with total proglacial river discharge

Total glacier melt calculated with the DDHM is compared with the discharge records estimated from the pressure sensor and the gauging records from the HydroChile station, at 500 m and 1700 m from the glacier snout, respectively, between 24 November, 2009 and 31 March 2010 (Figs. 1 and 11). At an hourly time step, glacier melt and proglacial discharge estimations have correlations of 0.72 (pressure sensor station) and 0.75 (HydroChile station). Melt estimated from the glacier represents between 42% and 58% of the streamflow estimated from the pressure sensor, depending on the ice $F_{DH}$ value used (Fig. 11). The remaining 42% to 58% of proglacial streamflow is attributed to contributions from glaciers and lakes in lateral valleys, but are not accounted for in the DDHM calculations. Moreover, during the first half of the season, the proglacial river includes snow melt runoff from the non-glaciated area of the valley.

Monthly melt from Universidad Glacier represents between 10% and 13% (depending on ice $F_{DH}$ used) of the total streamflow of the entire upper Tinguiririca Basin (1478 km$^2$) during the December 2009 to March 2010 period (Fig. 12, DGA station, Table 3). This percentage is much more than the area of the Universidad Glacier (~2%) as a portion of the total area of the upper Tinguiririca Basin. The percentage of glacier contribution is variable during the season (Table 3). At the beginning of the common period of pressure sensor and AWS1 measurements (end of November) streamflow is dominated by the snow melt across the entire upper Tinguiririca Basin. This is reflected in the high daily variability in streamflow at the DGA station until January, due to the control of air temperature over snow melt (Fig. 12). After the discharge peak at the end of January, the contribution of Universidad Glacier to total streamflow increased to 14-19%.

The daily variability of all stream gauging series was similar between December and January. The DGA station measurements mainly show the additional influence of the air temperature variations on snow melt across the catchment, since the rainfall in the period of Fig. 12 was 0 mm. In February and March, the DDHM calculated melt and the DGA station streamflow display similar temporal variations with one to two days of lag.

## 4 Discussion

### 4.1 Modelling approach and uncertainties

Our results suggest that a simple empirical melt model (DDHM) is suitable for estimating glacier melt contribution to streamflow from glaciers in the central region of Chile. This interpretation is based on the close correlation between melt estimates from the DHM and melt estimates from an energy balance model, ablation stake and sonic ranging sensor at a point scale, and agreement between estimates of total glacier runoff and discharge estimations in the proglacial stream. This good agreement results from: first, on-glacier measurements of meteorological data at two locations, enabling the use of a local hourly-calibrated lapse rate to extrapolate air temperature inputs to the distributed melt model; second, locally calibrated degree-hour factors; and third, knowledge of the spatial distribution of snow and ice cover from satellite data. Forcing distributed temperature-index melt models with off-glacier data can be problematic due to the difficulty in estimating the temperature distribution across the glacier (Shaw, 2017; Shaw et al., submitted). At a point scale, a locally-calibrated temperature-index model forced with off-glacier air temperature data can lead to improvement over use of on-glacier temperature data, due to damping of temperature within the glacier boundary layer (Guðmundsson et al., 2009). However, recent glacier studies have revealed high variability in the local air temperature lapse rate, due to variations in the strength and thickness of the katabatic boundary layer and changes associated with cloud cover and

synoptic-scale wind field (Petersen and Pellicciotti, 2011; Petersen et al., 2013; Ayala et al., 2015), which are difficult to account for in off-glacier data. Hence, the availability of temperature measurements for 2 on-glacier locations at different elevations provided suitable data for driving the DDHM.

Although we consider the model outputs to be robust, it is important to bear in mind that empirical temperature-index models do not attempt to simulate the real physical processes of glacier ablation, and the DDHM ignores other influences on rates and spatial patterns of ablation, such as topographic shading, blowing snow, debris-cover and subsurface fluxes. Hence, the DDHM may not be suitable for longer term mass balance studies where climatic and surface factors may undergo change.

The key sources of uncertainty in the results are: (a) The degree-hour factor of ice. A lack of stake measurements and only a short period of sonic ranger data on ice means there is some uncertainty in a representative ice $F_{DH}$ value at Universidad Glacier. A range of $F_{DH}$ values between 0.29 mm w.e. $°C^{-1}h^{-1}$ and 0.38 mm w.e. $°C^{-1}h^{-1}$ was applied to account for this uncertainty, but we note that published ice $F_{DD}$ values show a much greater range (Hock, 2003). Figure 7 shows that the accumulated melt of the DHM using an $F_{DH}$ of 0.29 mm w.e. $°C^{-1}h^{-1}$ was similar to the melt estimated by the EBM, stake and sonic range measurements in November. However, at the end of the comparison period (end of January), melt estimated using an $F_{DH}$ of 0.38 mm w.e. $°C^{-1}h^{-1}$ more closely match the energy balance melt estimation. This translates into an uncertainty of 11% in the cumulative runoff from the DDHM at the end of the ablation season (Fig. 11). As the streamflow at the DGA station is large, this ice melt uncertainty contributes only a small percentage of total streamflow (3%). (b) The snow line altitude derived from the MOD10A1 product. Although glacier surface characteristics on the tongue allow differentiation between ice and snow, the resolution of the snow product is similar to the width of the glacier tongue. A lag of 10 to 12 days was found between the MOD10A1 product and field observations of the transition from snow to ice at the AWS1 site (Fig. 7). Furthermore, in the highest zone of the glacier, fewer debris and aerosols cover the ice surface, making it harder to distinguish between ice and snow, which could have led to errors in identifying surface type. (c) DDHM melt estimates were not adjusted for the effects of moraine and patchy distributed debris in the ablation zone (Fig. 1). The moraines are of substantial thickness on lower areas of the tongue and likely to reduce ablation below the highest values shown in Fig. 8 in the terminus zone. However, other areas of the ablation zone are affected by a thin and patchy layer of debris or aerosol, which is likely to increase ablation through local albedo reduction (Fyffe et al., 2014). Although, quantification of the effects of debris on melt is beyond the scope of this study it would be expected that impacts of thick morainic debris and thin patchy debris elsewhere will tend to compensate in overall melt estimations for the glacier. (d) Snow density, which is required to convert stake and ultrasonic sensor measurements of snow into w.e. melt for model validation and calculation of degree-hour factors, was measured only two times in the early ablation period. (e) Single, fixed $z_0$ values from published literature were applied to snow and ice. Although the representativeness of these $z_0$ values cannot be evaluated due to lack of independent measurements, the small contribution of the turbulent fluxes to total melt means $z_0$ errors would only have a small influence on EBM output. (f) Sublimation was ignored in the DDHM. However, Universidad Glacier has an ablation regime dominated by melt, more typical of temperate glaciers further south in Chile (Brock et al., 2007), therefore this omission is likely to have led to only a small overestimate of glacier runoff. (g) Groundwater flows and evaporative losses from glacier melt water are unknown but considered negligible. (h) The date of the ASTER GDEM is not known, which could have produced small errors in temperature distribution due to elevation changes in the glacier surface between the dates of GDEM acquisition and model analyses.

During periods of low positive temperature and high insolation, the DHM tends underestimated melt, and *vice versa* during periods of high temperature, due to the high temperature sensitivity of simple temperature-index models (Pellicciotti et al., 2005). This implies spatial and temporal errors will occur, i.e. overestimation of melt during warm weather and on the lower glacier, and melt underestimation during cold weather and on the upper glacier. Such error will tend to compensate over time and in summation of total glacier melt, but will lead to short-term inaccuracies.

## 4.2 Glacier contribution to basin streamflow

The finding that Universidad Glacier, while accounting for just 2% of the total basin area, contributed a monthly mean between 10% and 13% of total streamflow from the entire upper Tinguiririca Basin over the December 2009 and March 2010 period, demonstrates the importance of glaciers for river flows in central Chile during the summer months. The overall glacier melt contribution to the Tinguiririca River would be much larger considering that the total glacier area of the basin is 81 km$^2$, representing 5.5% of the total basin area. Crucially, the glacier contribution becomes more significant over the course of the summer as other sources, principally the seasonal snowpack, deplete. Hence, glacier runoff becomes critical to maintaining flows in the Tinguiririca River during years when summer drought extends into autumn, e.g. in the period 2010-2015 (Bosier et al., 2016), and in dry winters when snowpack accumulation at high-elevation sites is small. Research by Gascoin et al. (2011) and Pourrier et al. (2014) on glaciers of the arid Andes has revealed the hydrological importance of glaciers to the north of the Tinguiririca Basin.

The recent and ongoing retreat of Universidad Glacier is a direct consequence of atmospheric warming (Le Quesne et al., 2009; Wilson et al., 2016) and the relevance of glacier melt contribution highlighted in this work, implies that serious negative impacts on river discharge are expected over the next decades. In the 1950-2007 period, a trend in runoff for the upper Tinguiririca Basin was observed (0.3 m$^3$ s$^{-1}$ y$^{-1}$, not significant) (Casassa et al., 2009). Considering that the estimated upward migration (200 m) of the zero degree isotherm between 1975 and 2001 in central Chile (Carrasco et al., 2005) far exceeds the elevational retreat (~60-70 m) of the Universidad Glacier snout (Wilson et al., 2016) over a longer period (1967-2015), the contributing melt area of Universidad Glacier has increased in the last ~30 years. Such increases in glacier melt might explain the positive discharge trends for several rivers in central Chile as suggested by Casassa et al. (2009). Another characteristic to consider is that the date marking the timing of the centre of mass of annual flow for the upper Tinguiririca River shows a negative trend in the period 1961-2007 (Cortez et al., 2011), indicating that the bulk of the annual flow is shifting towards earlier in the year. This implies that snowmelt occurs earlier and hence glacier ice is also exposed earlier in the year, increasing the hydrological importance of glaciers.

From our analyses, it is uncertain if the Tinguiririca River's discharge has already reached the "peak water" expected for glacierized basins as a consequence of deglaciation (Casassa et al., 2009). The observed recent positive trend in the discharge of the Tinguiririca River (Masiokas et al., 2006) suggests that peak water is yet to occur. In contrast, recent modelling work has shown that peak water has already passed further north in the Juncal Norte Basin and that future runoff is likely to sharply decrease (Ragettli et al., 2016). Estimations of the future runoff trend and melt contribution from Universidad Glacier are beyond the scope of this work. However, the possibility of increased persistence and recurrence of droughts in central Chile (Boisier et al., 2016) would increase the hydrological importance of Universidad Glacier in the future and therefore more research is needed in order to address these issues.

## 4.3 Comparison to other studies in Chile

It has been shown that on high altitude glaciers in northern Chile and in the dry season of the outer tropics of Peru and Bolivia, melt rates are reduced as more ablation occurs through sublimation (Winkler et al., 2009; Sagredo and Lowell, 2012; MacDonell et al., 2013), whereas, to the south of ~37° S, lower incident shortwave radiation and increased cloud cover reduce available energy for melt (Brock et al., 2007). Hence, Universidad Glacier may be located in a climatic zone which enhances high rates of summer melting as Sagredo and Lowell (2012) suggest in their climate zone classification for Andean glaciers. Local factors such as the large accumulation area and extension of the glacier tongue to a relatively low elevation also contributes to the high melt detected in the lower zone of the glacier (Fig. 8).

Although melt rates cannot be compared directly between different glaciers in different years, two other studies in Chile provide a context for the DDHM results for Universidad Glacier in the 2009-2010 season. Pellicciotti et al. (2014) estimated the total melt in the lower ablation zone of Juncal Norte Glacier (33°S) to be between 5000-6000 mm w.e. in the December 2008 to February 2009 period, which is slightly lower than the total melt of ~5000-6000 mm w.e. for the equivalent months of 2009-2010 at the AWS1 location on Universidad Glacier. Brock et al. (2007) estimated cumulative melt of 4950 mm w.e. and 3960 mm w.e in the January to March periods of 2004 and 2005, respectively, at 2000 m asl on Pichillancahue Glacier on Villarrica Volcano further to the south (39°S). This location likely represents the maximum ablation on Pichillancahue Glacier due to a continuous thick mantle of insulating tephra covering the glacier below this elevation. The total melt for the equivalent months of 2010 at the AWS1 location (2650 m asl) on Universidad glacier was higher, between 4800-5700 mm w.e., however, comparison of melt at different elevations on these two glaciers should be interpreted with caution.

Recently Ayala et al. (2016) found that the glacier melt contribution at the river outlet of glaciers Bello, Yeso (debris-free glaciers) and Piramide (debris-covered glacier) in the central Andes (~33.53°S), depends on the meteorological conditions of each year. In snow rich years, such as 2013-2014, glaciers contributed an estimated 30% of summer streamflow, while in dry years such as 2014-2015 the summer contribution was 50%. This latter value is similar to the glacier melt contribution recorded at the outlet of Universidad Glacier, which was in the range 42% to 58% of the total discharge estimated with the pressure sensor. Considering that almost no precipitation was recorded by weather stations close to the study site, the 2009-2010 ablation season is representative of relatively dry years in central Chile. However, as Ayala et al. (2016) suggest, melt contribution comparison between different glaciers must made with caution, considering that glacier melt depends of altitudinal range, glacier characteristics, differences in atmospheric conditions for each year and even differences in methodology.

At a basin scale, glacier contribution to downstream discharge in the Tinguiririca River is of similar magnitude to previous results for the central Andes. For example Ragettli and Pellicciotti (2012) estimated that 14% of the total streamflow of the Juncal River Basin (241 km$^2$, outlet at ~2250 m asl) was contributed by Juncal Norte Glacier (9.9 km$^2$) in the 2005/2006 hydrological year reaching a maximum of 47% during the late ablation season. For the Maipo Basin, Peña and Nazarala (1987) estimated a mean contribution from glaciers (~7.2% of the total upper Maipo basin area) of 11.8% between hydrological years 1981/1982 and 1985/1986, with maximum values towards the end of each hydrological year. An important issue raised in the results of Peña and Nazarala (1987) is the high interannual variability in the discharge from glaciers. For example, the percentage of the glacier contribution to total streamflow in the Maipo River in February 1983 was just 5%, but in February 1982 it was 34%. It has been suggested that another source of streamflow during the dry season is groundwater flow i.e. in the outer tropics in Peru, the groundwater contribution to outflow is greater than 24% in all of the analyzed valleys by Baraer et al. (2014). In central Chile, Rodriguez et al. (2016) estimated that contribution associate to subsurface storage in winter and fall is of 60% in Juncal Norte Basin. However it is difficult to estimate this contribution at Universidad Glacier without direct measurements.

## 5 Conclusions

In this study, we have investigated the meteorological conditions, ablation and melt water contribution to downstream river flow of Universidad Glacier, located in central Chile during the 2009-2010 summer ablation season. We used a point scale energy balance and a distributed degree-hour melt model, driven by data from two on-glacier weather stations. The main outcomes of this work are:

- The distributed degree-hour model provides a robust simulation of surface melt, especially on the glacier tongue where good agreement was found between melt estimated from the point scale degree-hour model, energy balance model, ablation stake

measurements and sonic ranger records. Almost continuously positive air temperatures in the ablation zone between November and March are well suited to the application of a simple temperature index method to calculate glacier melt, however, some melt overestimation was identified for the accumulation zones due to more frequent negative air temperatures at higher elevations.

- Meteorological conditions result in very high ablation season melt totals, which reach 10 m w. e. on the lower tongue. This finding is attributed to the high insolation due to a low percentage of cloud cover, combined with a predominantly positive air temperature.

- By comparing total glacier melt with discharge measurements at 50 km downstream on the Tinguiririca River, we estimate that the monthly mean contribution of Universidad Glacier is between 10% and 13% of the streamflow observed in the upper Tinguiririca Basin for the period December 2009 to March 2010. This estimated contribution reaches a maximum of 15% to 20% in March. The total contribution of all glaciers to streamflow in the upper Tinguiririca Basin will be considerably larger considering that Universidad Glacier only represents 36% of the total glacier area of the basin (~81 km$^2$).

The successful application of a simple temperature-index melt model to estimate total seasonal melt at Universidad Glacier is partly a consequence of the predominant high melt regime of this glacier, which favors the application of the degree-hour model. In this sense, estimation of streamflow contributions from glaciers in northern Chile is more challenging as an increasing proportion of ablation energy is consumed by sublimation (MacDonell et al., 2013) which cannot be estimated from simple temperature- index methods.

Climatic warming, leading to a rapid rise in the zero-degree isotherm (Carrasco et al. 2005) and upward expansion of glacier melt contributing area into the accumulation zone, means Universidad Glacier will continue to make a crucial, and perhaps an increasing contribution to downstream flows in the next few decades, particularly as smaller glaciers in the basin disappear. In the long term, glacier shrinkage will lead to a depletion of glacier melt and in downstream streamflow in the Tinguiririca River, particularly in late summer. This will have severe implications for human activities in the river valley such as mining, domestic consumption, industry, tourism, forestry and agriculture (Aitken et al., 2016) and hydropower generation (Valdés-Pineda et al., 2014). Hydropower generation on the Tinguiririca River at La Higuera and La Confluencia (Pelto, 2010) will be affected by interannual variability in water supply and future streamflow trends in the medium to long term. Finally, more long-term high elevation stations in the Andes are necessary to establish the inter-annual variability of glacier contribution to river discharge in order to help manage future water availability, considering climate change and the increasing demand for water in the region (Meza et al., 2012).

**Acknowledgments**

This work was supported by CECs, which is funded by the Chilean Government through the Centers of Excellence Base Financing Program of Comisión Nacional de Investigación Científica y Tecnológica de Chile (CONICYT). Pablo Zenteno and Camilo Rada assisted with data collection. The Dirección General de Aguas de Chile (DGA) also provided data and support to this paper. We would like to thank Christophe Kinnard for sharing his data. We would like to thank to two anonymous referees for their constructive and useful comments and recommendations.

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

**Table 1: Stake ablation measurements**

| Stake N° | Altitude (m asl) | Installation date | Measurement date | Difference (m) | Mean snow density (kg m⁻³) | Water equivalent (mm) |
|----------|------------------|-------------------|------------------|----------------|----------------------------|-----------------------|
| 1 | 2646 | 30-09-2009 | 21-11-2009 | -1.23 | 422 | 519 |
| 2 | 2828 | 02-10-2009 | 21-11-2009 | -0.81 | 441 | 357 |
| 3 | 2939 | 03-10-2009 | 21-11-2009 | -0.33 | 413 | 136 |

**Table 2: Mean monthly energy fluxes at AWS1**

| | Incoming shortwave radiation [W m⁻²] | Net radiation [W m⁻²] | Latent heat [W m⁻²] | Sensible heat [W m⁻²] | Melt energy [W m⁻²] |
|---|---|---|---|---|---|
| October 2009 | 238 | 43 | -43 | 18 | 17 |
| November 2009 | 279 | 99 | -28 | 16 | 87 |
| December 2009 | 373 | 249 | -13 | 19 | 255 |
| January 2010 | 322 | 225 | -6 | 30 | 249 |

**Table 3: Monthly discharge from Universidad Glacier as percentage of the total discharge in the Tinguiririca River, measured at the DGA station. Ranges in the percentages are for $F_{DH}$ ice values of 0.29 mm w.e. °C⁻¹ h⁻¹ and 0.38 mm w.e. °C⁻¹ h⁻¹.**

| Months | Monthly Mean |
|--------|--------------|
| Dec-09 | 4.3% - 5.2% |
| Jan-10 | 8.1% - 10.2% |
| Feb-10 | 14.1% - 17.9% |
| Mar-10 | 15.3% - 19.5% |
| Mean of the period | 10.5% - 13.2% |

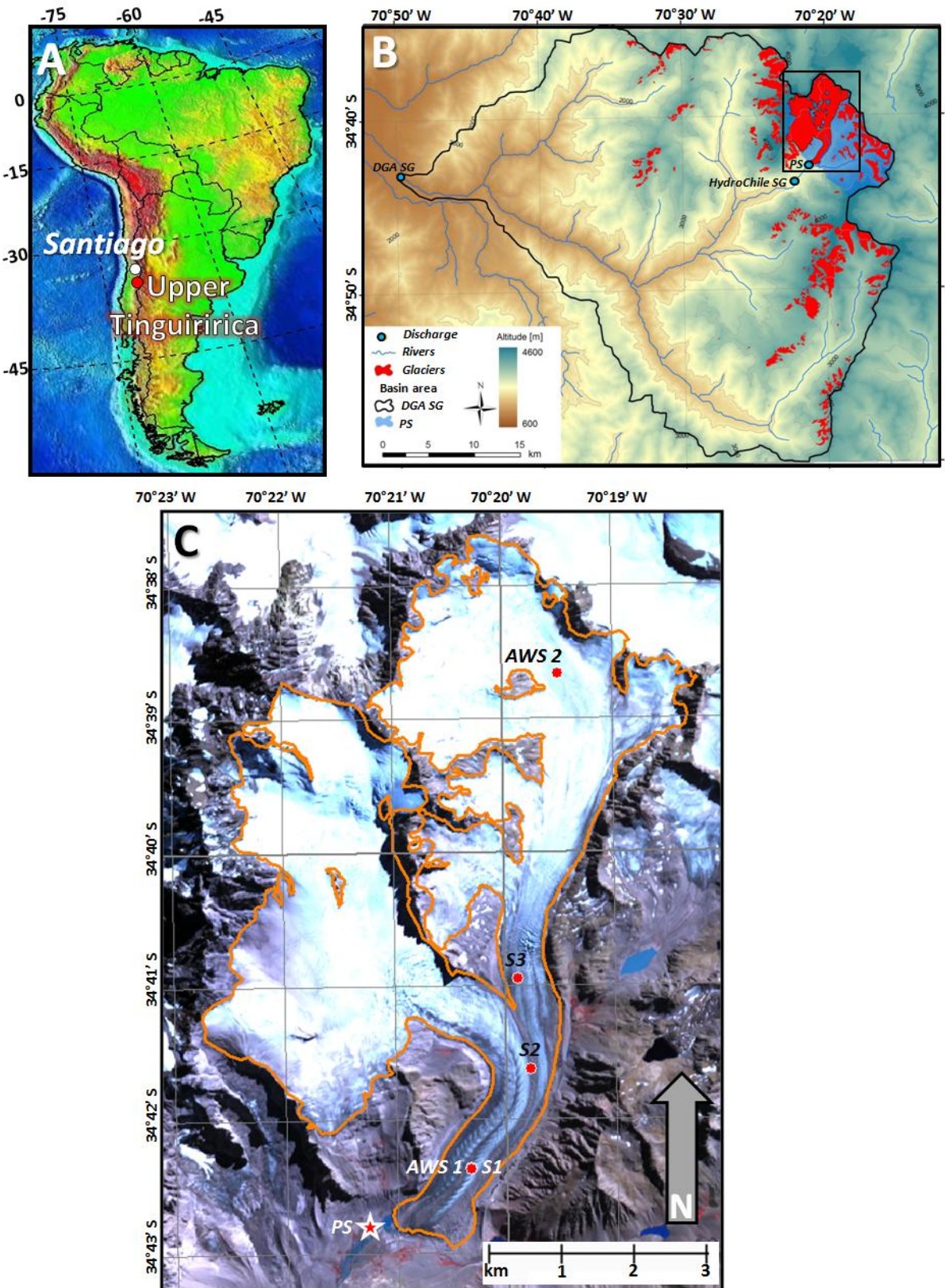

**Figure 1: Location of Universidad Glacier in central Chile. Panel A shows the regional location, panel B shows the upper Tinguiririca Basin and panel C shows Universidad Glacier (orange outline), automatic weather stations (AWS) and ablation stakes (S) installed. PS indicates the location of the pressure sensor, SG indicates the locations of the DGA and HydroChile stream gauge. The background is an ASTER image from 27 March 2010, UTM 19S.**

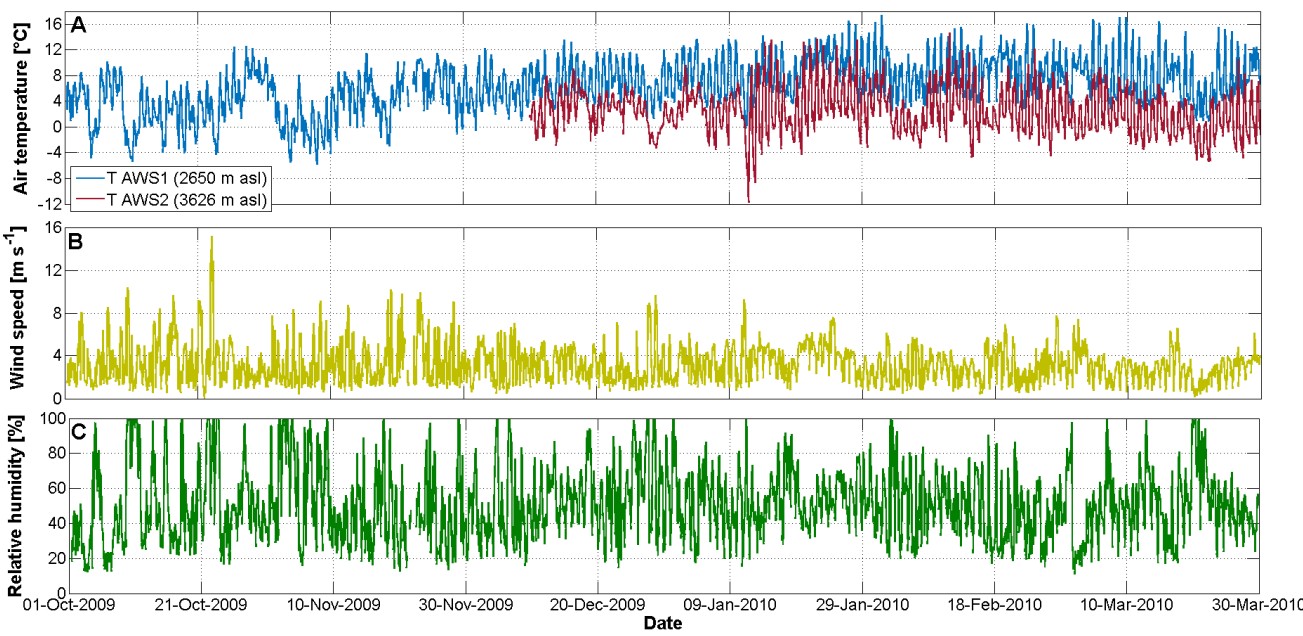

**Figure 2: Hourly time series of observed meteorological variables. A) Air temperature at AWS1 and AWS2, B) Wind speed at AWS1 and C) Relative humidity at AWS1.**

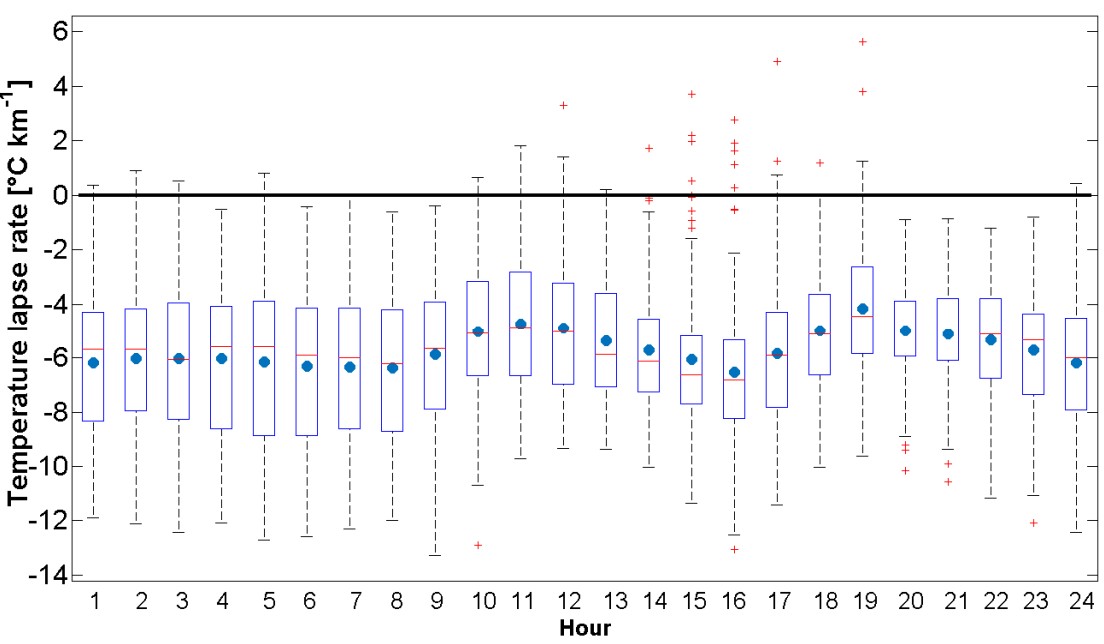

**Figure 3: Boxplot showing the statistical distribution of hourly lapse rates calculated between AWS1 and AWS2 in the common period. Upper and lower box limits are the 75% and 25% quartiles, the red horizontal line is the median, the filled circle is the mean, and crosses are outlying values.**

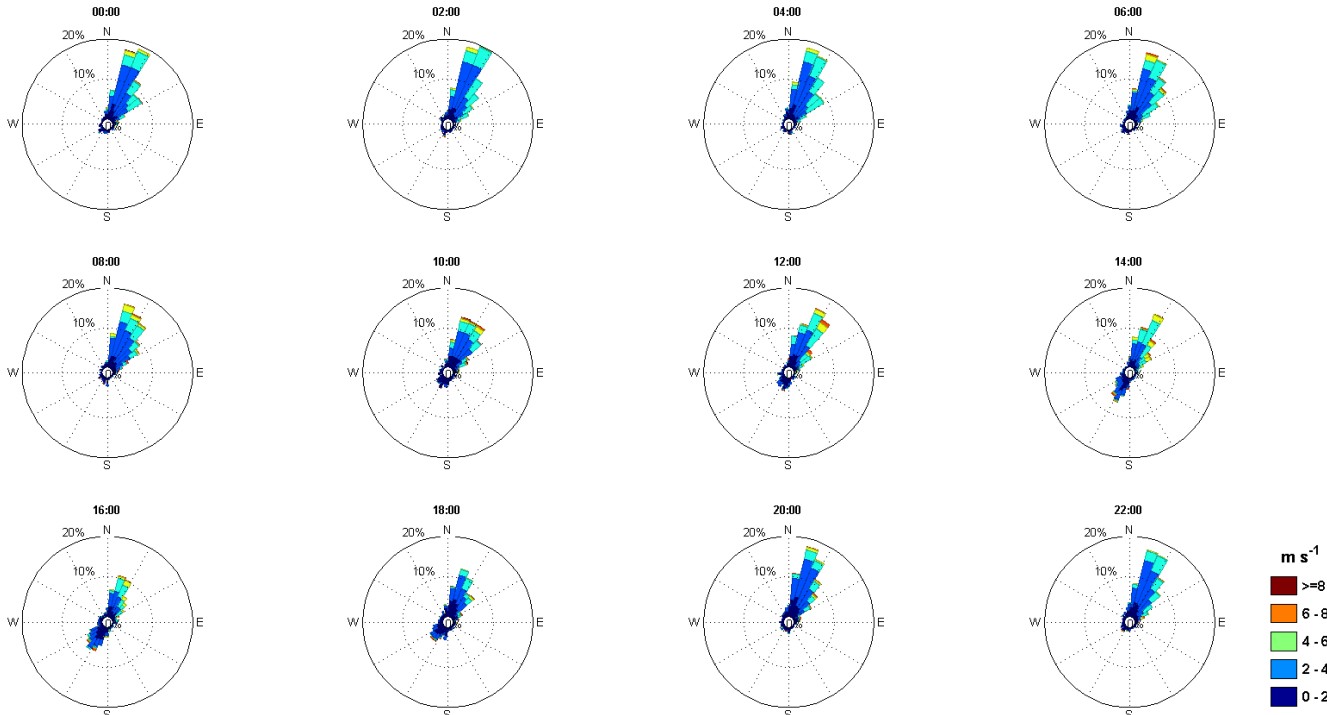

**Figure 4: Wind roses showing the hourly wind direction and the wind speed frequency at AWS1 (local time).**

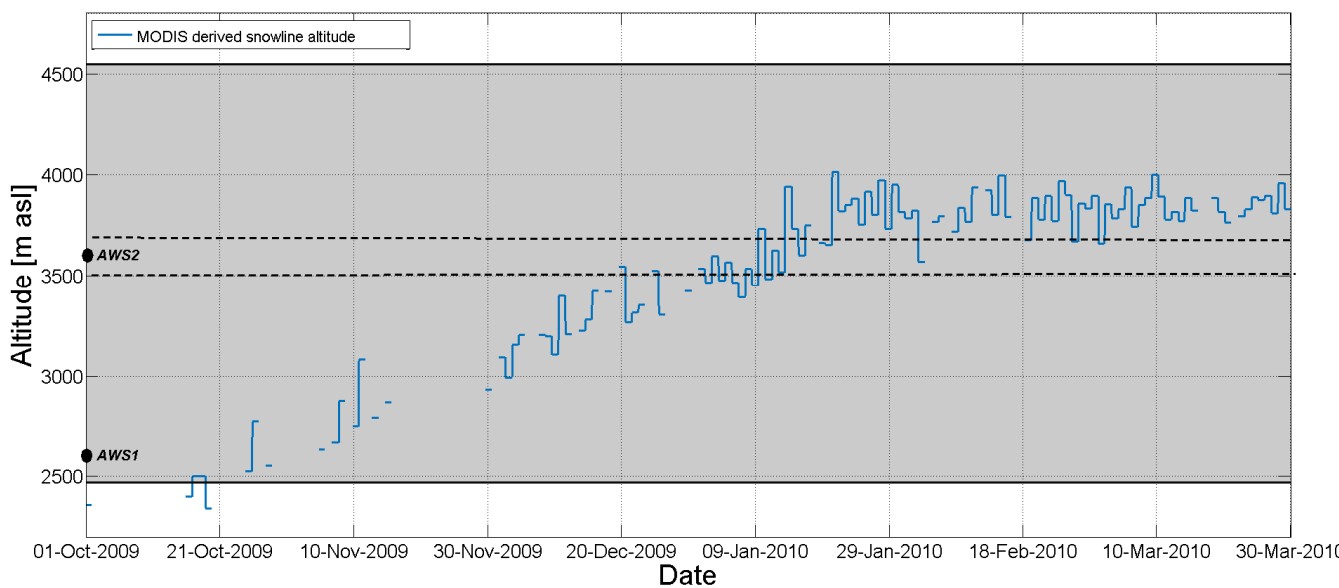

5    **Figure 5: Snow line elevation estimated using the MODIS snow cover product. The grey area corresponds to the altitude range of Universidad Glacier, the dashed line shows the equilibrium line altitude range estimated using an ASTER image of 27 March 2010 and black points show the AWS elevations.**

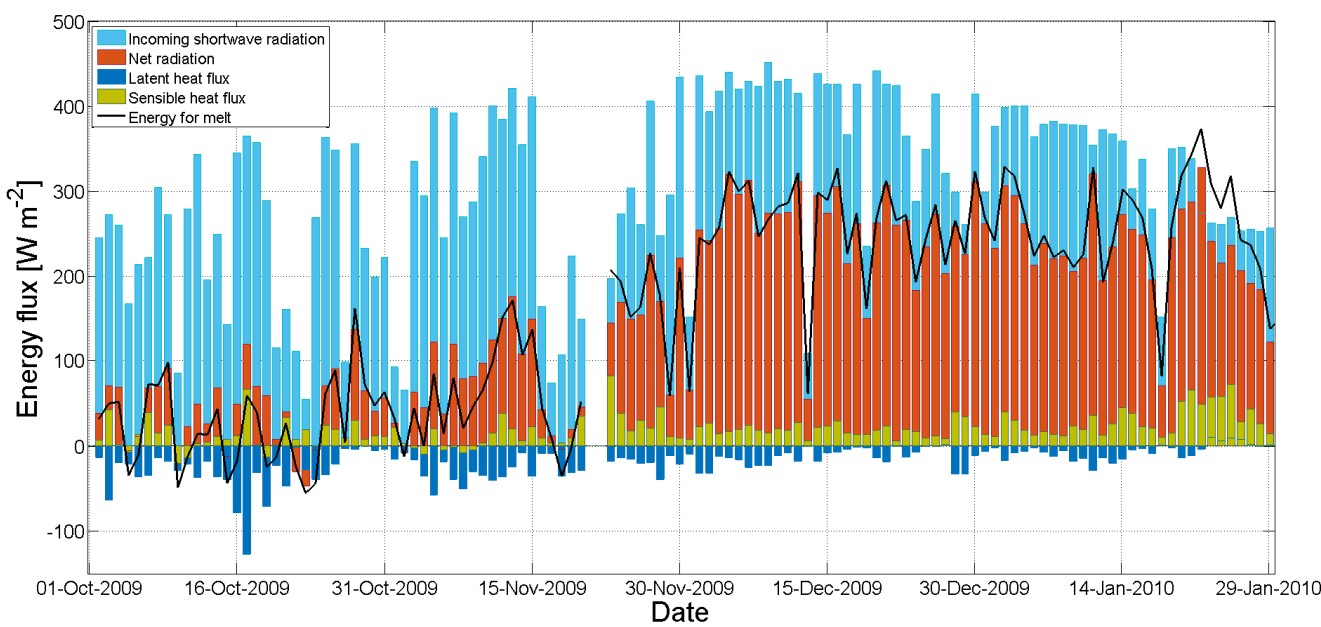

**Figure 6: Daily mean net radiation, incoming shortwave radiation, latent and sensible heat fluxes and the calculated energy available for melt at AWS1 (2650 m asl). On 21 and 22 November there are no data due to maintenance of AWS1.**

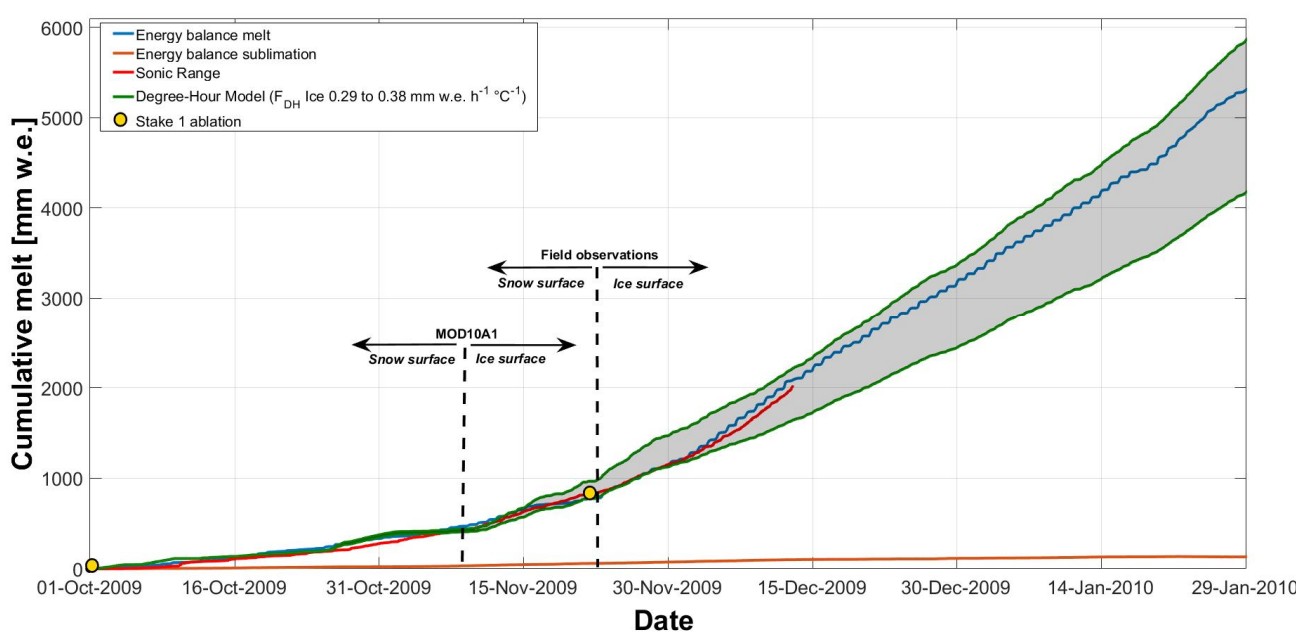

**Figure 7: Comparison of cumulative melt estimated by the point scale degree-hour model (grey area between green lines), point scale energy balance model, sonic ranger and stake 1 located near AWS1 (2650 m asl).**

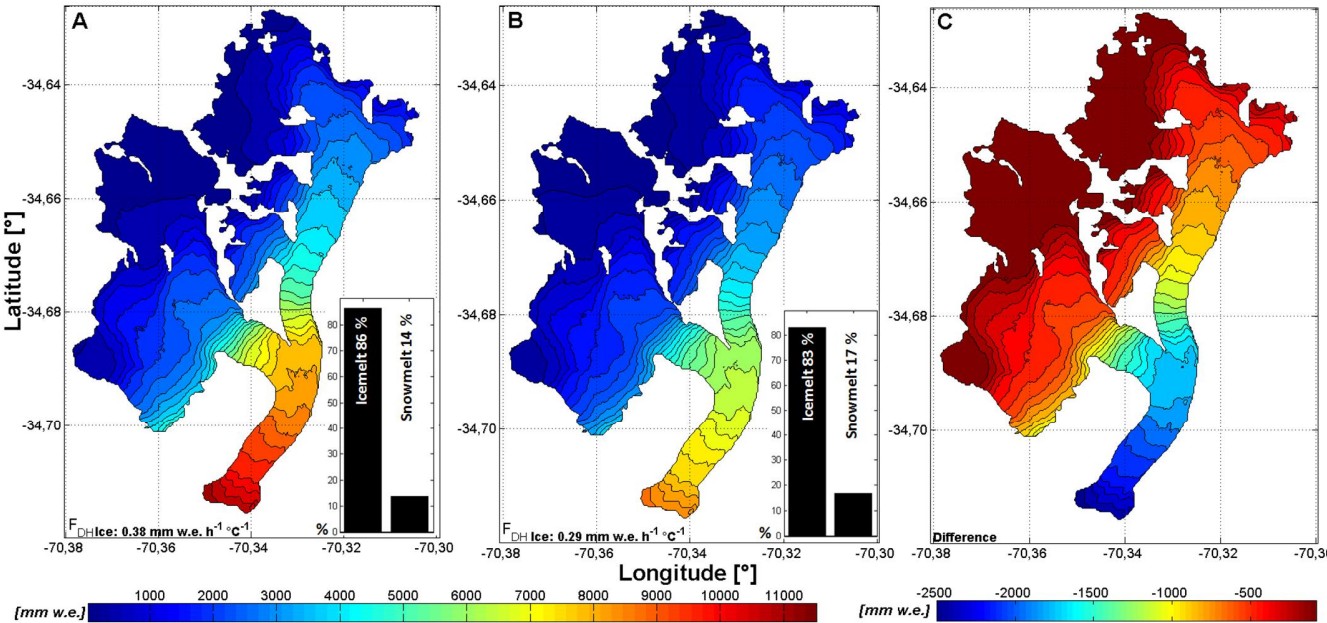

**Figure 8: Spatial distribution of cumulative glacier melt for Universidad Glacier using two different $F_{DH}$ values for ice. A) $F_{DH}$ = 0.38 mm w.e. h$^{-1}$ $^{\circ}$C$^{-1}$, B) $F_{DH}$ = 0.29 mm w.e. h$^{-1}$ $^{\circ}$C$^{-1}$ and C) Difference of panels A and B. Totals are for the October 2009 to March 2010 period. In panels A) and B) the percentage of contributions of snow and ice surfaces to total melt are shown.**

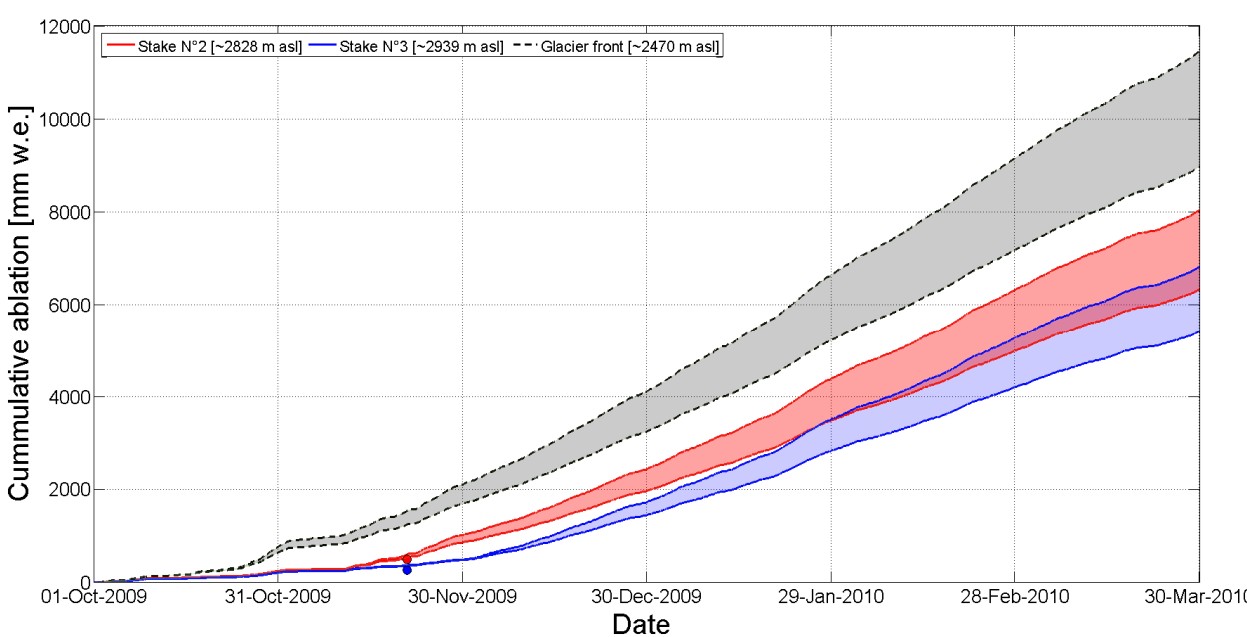

**Figure 9: Total cumulative melt of Universidad Glacier using the degree-hour model. The red and blue lines and areas represent the cumulative melt at the locations of stakes 2 and 3, respectively. Points indicate the stake measurements. The area in grey enclosed by dashed black lines represents the lowest altitude of the glacier.**

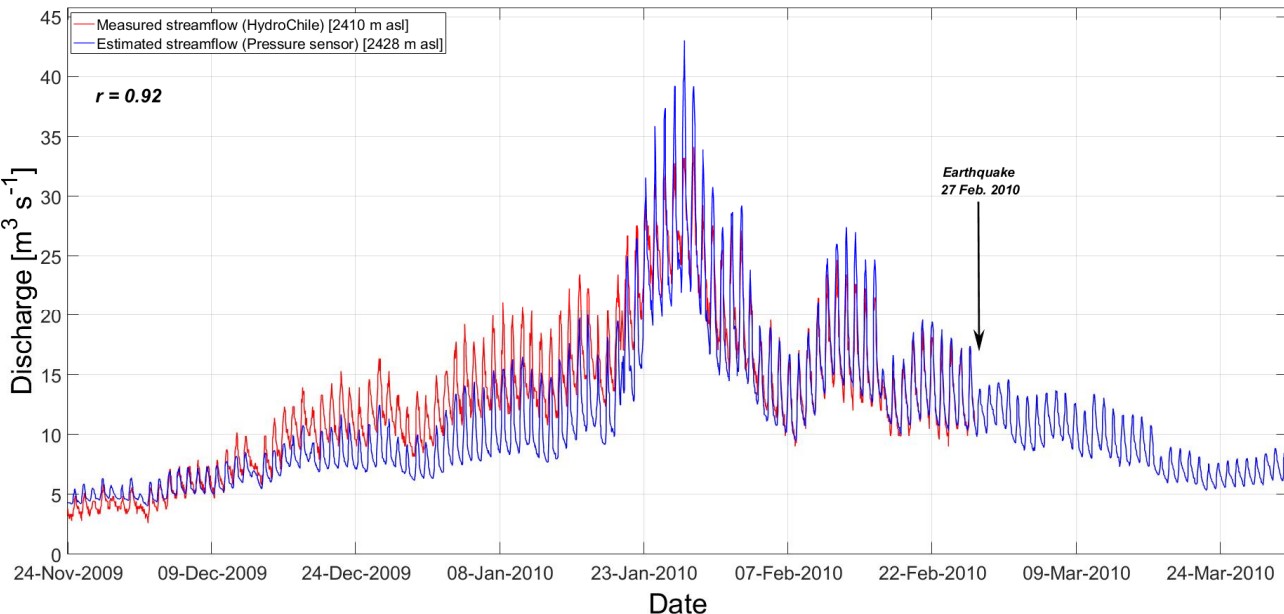

**Figure 10: Time series of hourly discharge in the proglacial stream from the water level pressure sensor and the HydroChile gauging station.**

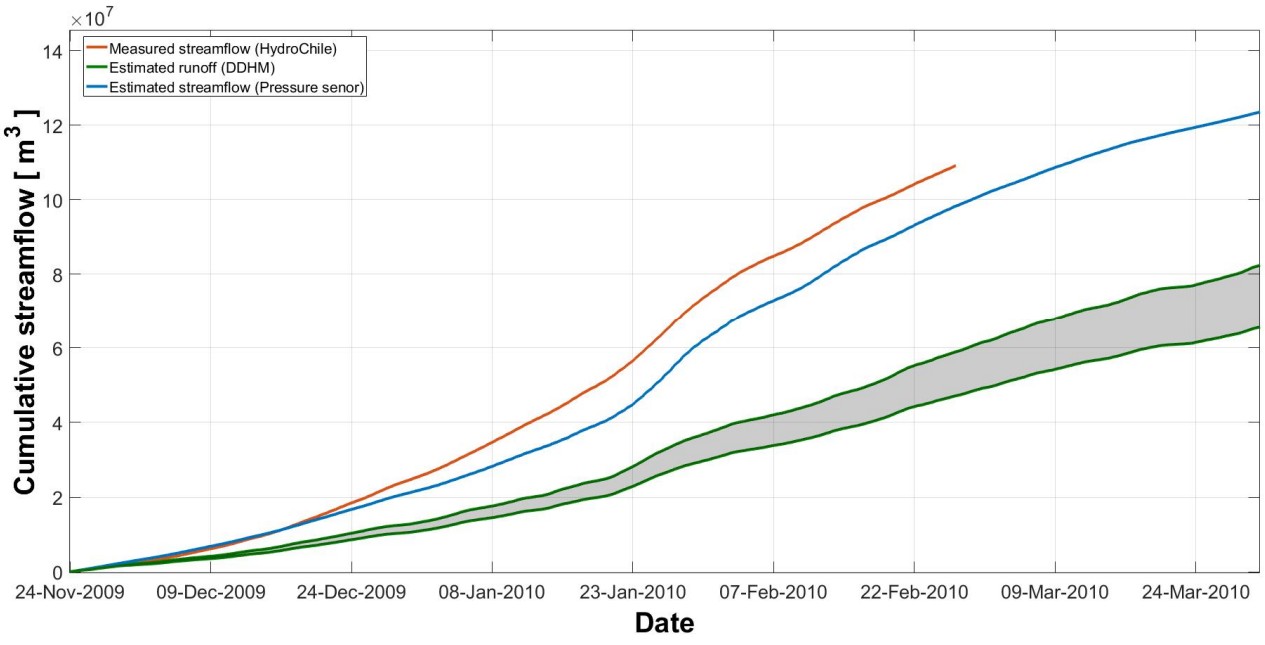

**Figure 11: Comparison of cumulative melt calculated with distributed degree-hour model (grey area), and streamflow measurements from the water level sensor data and the HydroChile station.**

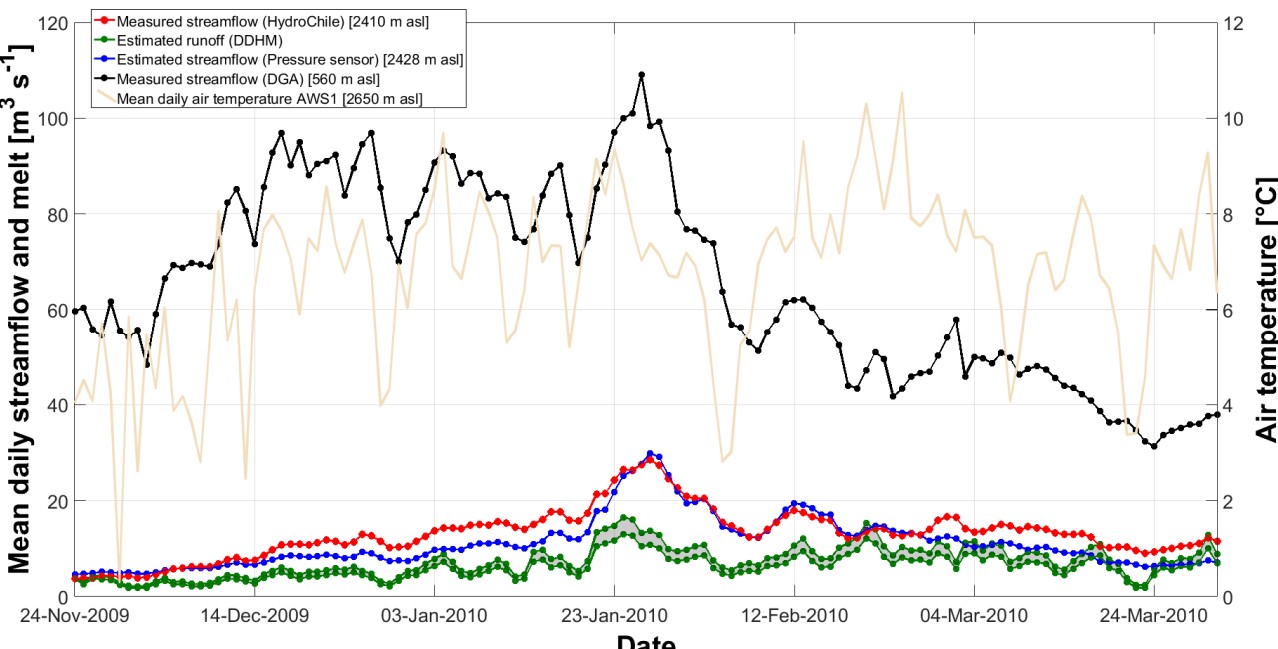

**Figure 12: Daily mean melt from the distributed degree-hour model, and discharge measurements from the water level sensor, HydroChile station and DGA station. Mean daily air temperature at AWS1 is plotted on the right y axis.**

