# Peer review of "Assessing glacier melt contribution to streamflow at Universidad Glacier, central Andes of Chile"

_Hydrology and Earth System Sciences, 2016_

## Referee Comment (RC1) · Anonymous Referee #1 · 17 Nov 2016

General comments: The authors present an important contribution of glacial hydrology in the Andes of central Chile using both direct in-situ measurements and remote sensing data in order to force satisfactorily a distributed degree-hour model and estimate glacial melt contribution to river runoff in the ablation period 2009-2010.

While the scientific significance is high regarding the research lack of information of insitu glacier streamflow contribution measurements and the methods are well-applied, the conclusion/discussion and scope of the research should be improved. Printer-friendly version

The authors must contextualize their findings with more studies about glacial hydrology, discussing more thoroughly the current status of Universidad glacier (area decrease of the last decades?) and its possible future point of peak water considering current shrinkage rates as well as local and downstream impacts of changing river runoff (in the region). It is not clear, if the results should be seen as a first short snapshot (only 5-6 months measurements) at the beginning of an anomalous period of drought (2010-2015) or if they can be brought into a wider context (ideally with longer in-situ data). While relative glacier melt fraction to river runoff might be high particularly in dry periods and the upper Tinguiririca catchment, relative contribution is expected to decrease with increasing distance from headwaters, i. e. for the low-lying coastal cities and water users. The mention of (the insignificance of) groundwater flows, probably difficult to estimate without direct measurements / tracing methods, should be revised as many different hydrological models have not been capable to adequately represent groundwater flows. Some studies of the last years suggest that they represent an important driver (e. g. Baraer et al., 2014 for the outer tropical Andes of Peru).

Furthermore, several typing and grammatical errors and imprecisions can be identified. For publication, English (errors, vocabulary, redundancies) should be improved.

The manuscript contains multiple tables and figures, most of them helpful for further comprehension, others less substantial. In order to reduce total paper volume, I would skip e.g. Table 1 and Figure 4. However, all anomalies / data gaps in the plots should be briefly indicated and explained in the text or subtitles.

In summary, I recommend a thorough minor revision.

Specific comments: 1 / 10-11: is that true that glacier melt represents more than the half of total streamflow contribution in lowlands during dry years in Chile? I would rather expect a reduction of relative contribution with increasing distance from the glacier and headwaters converting glacier streamflow to an important but not the main contributor in the lowlands. 2/38 - 3/1: what about glacier area and (estimated) volume changes
and current retreat rates of Universidad glacier and/or in the region? 3 / 7-8: you identify the year 2009/2010 as (just) the beginning of a longer dry season (2010-2015) but it is unclear why you did not incorporate a longer period of measurements into your study 3 / 18-19: again, you do not explain why your study only covers six months of data measurements 3 / 33-34: how did you discriminate snow from ice with the NDSI? Thresholds and techniques should be mentioned 4 / 3: clarify which images were selected with a cloud cover threshold: Landsat 5 TM? 4 / 14-15: the explanation of how to convert hourly to daily format is very basic and can be neglected 10 / 29-31: again, be careful that you distinguish upstream from downstream (lowland) glacier streamflow contribution, the latter possibly less significant; what about flow contribution in austral winter? Although you have only worked in the ablation period, it would be good that the reader gets a general idea of glacier streamflow contribution changes during a whole hydrological year 11 / 3-6: the point of (future) peak water is not sufficiently investigated in many mountainous regions worldwide but an increasingly important research question, particularly for future water management, can you examine this guestion about the possible peak water of Central-Andean glaciers in Chile a bit more? More literature? 11 / 13-14: is it true that melt rates are generally reduced further north (until where?) of Universidad glacier? Sublimation process are strongest with a pronounced water vapor gradient which is true for the dry season of e.g. the outer tropics (Peru/Bolivia) but not for glaciers in the inner tropics. 11 / 37 - 12 / 1: is Universidad glacier really such a particular glacier with highest melt rates in Chile? cite comparing literature 12 / 2-3: this affirmation is obsolete as it represents a typical mechanism of glacier energy budget and mass balance 12 / 15: does groundwater flow really become depleted? any studies (e. g. tracers: Rodriguez et al., 2014)? in other parts of the Andes (where reduced ablation also takes place during the winter season) groundwater has been identified to be a strong contributor and generally underestimated in many studies 12 / 24: the last argument should be more developed. The region is important for multiple water users. As an example, just some kilometers downstream, the hydropower plants La Higuera / La Confluencia are situated and possibly strongly

**HESSD**
affected by annual/seasonal changes in river runoff 16 / Table 1: this table does not contribute substantially to the study comprehension, therefore I would take it out 18 / Table 3: indicate period in the title "(2009-2010) 20 / Figure 1: upper left: the three gauges are not clearly identifiable; the map text "CECs HydroChile" confuses; also, the abbreviations "CECs" and "DGA" in the legend are not proper; text of the figure: add "(orange outline)" after "Universidad glacier" 21 / Figures 2-13: indicate altitude (m asl) for ALL station data 23 / Figure 4: in order to reduce paper volume, I would skip this graph as it does not substantially contribute for a further process comprehension 25 / Figure 6: eliminate "[dd-mmm-yyyy]" at x-axis legend; you also do not use this definition in Figure 7 26 / Figure 7: indicate gaps which are present between November 21-22 31 / Figure 12: no runoff measurements from March on? explain this data gap

Technical corrections: 1 / 1-3: with 28 words, the title is too long and complicated. A more concise title would be: "Glacier melt contribution to river runoff at Universidad glacier, central Andes of Chile" 1 / 11: eliminate "the" before "glacier melt" 1 / 13: insert "within the" before "central Andes of Chile" 1 / 19: replace "altitude part" by "ablation area" 1 / 28: insert "a" before "crucial resource" 2 / 21: change order "Mediterranean climate type" 2 / 26: use directly the previously introduced abbreviation "AWS" 2 / 32: correct "altitudinal range" 2 / 33: improve "which converge at an altitude" 2 / 35-37: change order considering a clockwise aspect of glaciers (north to the west) 3 / 2-3: "fastest period" does not exist, improve 3 / 9-10: three times the word "measurements", replace 3 / 14-15: not a full phrase, a verb is missing! 3 / 16: correct "net all-wave radiation" 3 / 32: insert "spatial" before "resolution" (there are also other types of resolutions) 3 / 33: better specify "Landsat 5 TM (30 m spatial resolution)" 4 / 1-2: eliminate the long parenthesis "(Advanced Spaceborne... Version 2)" 4 / 30-31: improve phrase: it is not "melt overestimation" which is dominated by melt from the ablation zone; instead of "however" you could use "as it" 5 / 5: "and the afternoon maximum" could be the beginning of a new phrase and needs a verb 5 / 14: include "shortwave" before "radiation" 5 / 21: insert "to be" before "a constant" 5 / 22: add "a" before "function" 8 / 2: eliminate "(100% relative humidity" - very basic 8 / 4: correct "was covered" 8 / 21-22:
improve phrase, you could separate it into two phrases from "fluxes calculated by" on inserting a new verb 10 / 3: eliminate "a" before "suitable" 10 / 4: use also the word "correlation" instead of only "agreement" 10 / 8: improve, e. g. "an hourly calibrated lapse rate at the glacier" 10 / 19: maintain the same terms, here "Universidad glacier" 10 / 37: replace "that" by "than" 11 / 14: add "cover" after "cloud" 11 / 20: eliminate "the" before "each" 11 / 24: add "the" before "central Andes" 11 / 25: correct "depends on"; insert "as" before "2013" 11 / 26: correct phrase "while in dry years" 11 / 28: better write "climatic conditions" instead of "meteorology" 11 / 29: insert "model" after "melt" 11 / 34: a final point is missing before "The ablation" 11 / 37: improve phrase avoiding the semicolon with e.g. "and are thus greater" 12 / 2: "latitudinal" instead of "latitude" 12 / 3: "persistent" instead of "persistence" 12 / 11: add "km" after "1.7" 12 / 12: a space is missing before "The total" 12 / 13: improve phrase: "Universidad glacier only represents 36%" 12 / 14: add the year "2010" after "March" 12 / 20: insert "the" before "zero-degree"

References: Baraer, M, J. Mckenzie, B. G. Mark, R. P. Gordon, T. Condom, J. T. Bury, J. Gomez, S. Knox, and S. Fortner, "Contribution of groundwater to the outflow from ungauged glacierized catchments: a multi-site study in the tropical," Hydrol. Process., 2014.

Rodriguez, M., Ohlander, N. and McPhee, N.: Estimating glacier and snowmelt contributions to stream flow in a Central Andes catchment in Chile using natural tracers, HESS Discussion papers, 2014.

---

## Referee Comment (RC2) · Anonymous Referee #2 · 7 Dec 2016

Review for Hydrology and Earth System Sciences

**Title: Assessing glacier contribution to river runoff in the Andes of central Chile: Analysis of in situ weather station data, runoff measurements and melt modelling at Universidad glacier (34° 40'S, 70°20' W)**

Authors: Bravo, Loriaux, Rivera, Brock

**PAPER SUMMARY AND RECOMMENDATION**

Bravo et al. estimate the runoff contribution of Universidad Glacier (~34°, central Chile) during the austral summer 2009-10. The authors use a set of meteorological, glaciological and hydrological measurements to run a point-scale energy balance model and to calibrate a degree-hour model, which is later used to calculate melt at the distributed scale. The authors found that glacier melt rates are extremely high (>10 m w.e.) at the glacier terminus and that the runoff contribution of the glacier represents a 10-13% of the summer runoff of the upper Tinguiririca River catchment (560 m asl). This contribution reaches 34% during late summer (March). They also conclude that a temperature-index model provides good estimations due to the availability of on-glacier data and the observed large melt rates.

The manuscript is within the scope of HESS, it is in general well-written and most of the methodology is clearly explained (with some exceptions). The discussion section needs to be strengthened and the conclusions are somehow vague. I think this paper provides interesting data and results to the glaciological and hydrological communities and, in my opinion, it will be acceptable for publication after solving/clarifying some issues (see major comments) and a careful revision of the text (see minor comments).

**MAJOR COMMENTS**

*1. Specify if the glacier contribution to runoff is coming from the snow or the bare ice.*

As the authors use MODIS images to decide the type of surface of each grid cell, it would be easy to calculate the runoff contribution from the snow and the bare ice separately (as in (Huss, 2011; Ragettli and Pellicciotti, 2012; Ayala *et al.*, 2016)). Melt rates of bare ice are important to estimate (maybe in future studies) the reduction of ice volume.

*2. Validation of the degree-hour method in Figure 6*

The authors justify the use of a temperature-index model based on the good agreement with the EB model results and the ablation measurements (Figure 6), but, why the time period in the x-axis of Figure 6 finishes on December 12? I think that the authors have the necessary data to run the EB and the degree-hour models after that date. I know that the sonic range stops on December 12, but it would be interesting to see the comparison with the EB model. Maybe I missed something, please clarify.

*3. Comparison to other studies (section 4.3)*

In order to stablish a meaningful comparison with other studies, previous results from other catchments should be always provided with: i) the percentage of glacierized area, ii) the elevation of the catchment outlet, iii) the analyzed time period and iv) if they refer to ice melt or glacier (i.e. including snow and ice) melt. Otherwise it is difficult to extract clear conclusions. For example, in lines 11/10-13, the authors mention that Brock et al. (2009) estimated 3960 to 4950 mm w.e. on Pichillancahue Glacier. The authors state that the ablation measured on Universidad Glacier doubles

that of Pichillancahue Glacier, but the time periods covered by these studies are very different in extension (Pichillancahue from January to March, Universidad from October to March).

**4. *DHF for snow**

I think there might be an underestimation of the DHFs for snow (lines 4/11-15). If total ablation at S1 is 519 mm w.e. (Table 2) and the average positive air temperature is 3.5°C, is the DHF of 0.12 mm w.e. h-1 °C-1 coming from the use of all time steps (52 days*24 hours) in equation (1)? Shouldn't you use only the time steps with positive air temperatures? What is the percentage of time steps with negative temperatures before the 21[st] of November? Please briefly explain the mentioned procedure of Braithwaite et al. (1998).

**5. *Text corrections**
- The introduction would benefit from a couple of recent articles by (Mernild *et al.*, 2015, 2016).
- Section 2.5 (temperature-index models) should be written more clearly.
- Please improve the content and structure of the conclusions section. I suggest enumerating the main outcomes of the study.
- Please perform a careful proofreading of the entire article.

**MINOR COMMENTS**

- Please correct systematically throughout the paper the use of capital letters for glaciers and rivers. Universidad glacier -> Universidad Glacier, Tinguiririca river -> Tinguiririca River.
- Check the use of present and past tenses. You frequently change from one tense to another, especially in the methods section.

1/11: "during late summer and autumn"

1/12: "To address these shortcomings", why in plural? You only describe the fact that few studies are available.

1/15: "to compare"

1/28: "water is a crucial resource"

1/28-29: Could you add more references apart from Masiokas et al. 2006 to sustain this sentence? The study of Masiokas et al. 2006 is about snowpack variations and not water uses.

1/30-31: Please reword: "In this region, winter precipitation is driven by the interactions…., and summer runoff (or summer water supply) by the storage and release from glaciers and the seasonal snow cover".

1/32: Maybe replace "water supply" by "runoff generation".

1/35: I think you need to replace "altitude" by "elevation" if you refer to terrain. How do you calculate the 4000 m asl? Is it the average of the peaks? Do you have a reference?

2/2: "warm temperatures"

Consider to change "trigger" by "produce" or "cause"

2/3: "rivers in the Andean basins of central Chile are mainly driven by the melting of the seasonal snowpack."

Please specify that you refer to the highest river sections. The annual regime is driven by winter precipitation in the lower sections.

2/3-4: The expression "is related to the existence" sounds awkward.

2/6-7: "For example, Peña and Nazarala (1987) estimated that the contribution of glacier melt to the Maipo River basin in summer 1981/82 was maximum in February and represented 34% of total discharge".

Please provide the elevation of the outlet and the percentage of the glacierized area of the catchment analyzed by Peña and Nazarala (2012). Also if by "glacier melt" they include the seasonal snow over the glacier.

2/10-12: Please provide one or two sentences with the main conclusions of Pellicciotti et al. (2008). Otherwise this reference is not very meaningful.

2/12-14: Please provide the elevation of the outlet and the percentage of the glacierized area of the catchment analyzed by Ragettli and Pellicciotti (2012). Also if by "glacier melt" they include the seasonal snow over the glacier.

2/14-16: Please check this sentence. "Results are available only for one basin" sounds strange.

2/17: "or on the impact of…"

2/18-19: can you be more specific? What do you mean by "melt patterns"? Temporal, spatial?

2/20: deficiencies -> knowledge gaps, issues

2/33: the words "which convergent" are not clear.

From where did you obtained the ELA? References?

2/33-34: Below this -> Below this elevation, below the ELA

3/10: "After the analysis of energy fluxes at the location of the lower AWS, a temperature-index model was calibrated and applied at the glacier scale. Resulting melt amounts were used to estimate total glacier discharge, which is compared with downstream discharge records."

3/21: Please choose another title. Snow density is not an ablation measurement.

3/22: re-measured -> read

3/26: melt -> of surface ablation. The ablation stake also includes sublimation.

3/27: "(Table 1). The sensor recorded surface …".

3/31-32; Please provide more details about the regression between Modis and Landsat products. At least the basic principles.

3/35: elevational distribution of snow cover ->  snow line

4/1: What is the acquisition date of the DEM? Is it similar to that of the study period?

4/3: "Images were used…" , Modis images?

4/3-4: Remove "For modelling purposes".

4/8: remove the second "applied".

4/10-11: ", which we refer to as the degree-day factor, …".

4/11-12: stake 1 melt measurements -> stake 1 ablation measurements.

4/11-13: This sentence is not clear. Please reword, perhaps you should split it in two.

4/12: "negative temperatures are set to 0°C" This sounds very strange. Say instead that you set to zero all melt occurring at time steps when the air temperature is below the temperature threshold.

4/13: Please place the value 3.5°C in another part of the sentence.

4/13 "With these values" What values?

4/13: "Following the procedure of Braithwaite et al. (1998)" What procedure? Please briefly explain it. Do you divide the total ablation by the total number of hours or only by those with positive air temperatures?

4/14: Why would you multiply by 24? I would think that melt only occurs during daytime (maybe 14 or 16 hours per day).

4/16: Please add at the beginning of the sentence a short explanation of why you cannot use the same procedure as for snow: "As we do not have ablation stake measurements in the period when the ice surface is exposed, we use a range of published….".

4/21: Can you use only one symbol? Either Df or DHF.

5/3: Since these are negative values, maybe write "with a minima in magnitude".

5/6: It should be "entrainment of warm air from the upper atmospheric layers". Please see the articles from van den Broeke (1997a, 1997b) in Pasterze Glacier for a more theoretical perspective. Insolated bare rock surfaces can also locally increase near-surface air temperature, but I don't think that "entrainment" is the right term.

5/6-7: Could you please check if wind directions reveal up-winds from the proglacial valley? Petersen and Pellicciotti (2012) observed this feature in Juncal Norte Glacier.

5/12: "was determined following Oerlemens (2010)". Remove also the parenthesis.

5/17: Why do you need the reference of Oerlemans and Klok to neglect the heat from the rain? Or is the reference wrongly placed?

5/19: "The sensible heat fluxes were calculated…"

5/22-23: Do you assume the same value of z0 for snow and ice?

6/3-5: I guess this is ok, but you are assuming the surface temperature as 0°C for the sensible heat fluxes, so, to be consistent, everything should be evaporation.

6/13: You missed the evaporation rate.

6/26: Do not mention what you did not do, delete "There were no direct measurements…"

6/28: Add a space before "Water level…"

7/30: replace "almost always" and "more frequently" by a percentage of time.

8/3-6: Can you say something about the ELA with this procedure? If we use the elevation of the snowline at the end of the ablation season as an indicator of the ELA that year, we would get a number much higher than the value of 2900 m asl (mentioned in line 2/33).

8/14: Please use the same number of significant digits for the DHFs (in lines 4/18 you use only two).

8/18-19: Please see main comment number 2.

8/32-33: Move to methods.

9/3: October -> November

9/15: "purposes".

9/19-20: Do you have data before November 24? Why don't you start the comparison on October 1?

9/23: "…contributions from glaciers…"

9/23: Please mention these lakes in the catchment description.

9/29-31: Please explain this sentence better:

"At the beginning of the common period": What period do you mean exactly?

"in the basin": what basin? The largest one?

Why is the high daily variability associated with the control of air temperature over snowmelt?

10/3: "is suitable"

10/4: "high melt regime" is not a very precise term. Do you mean something like "large retreat during last years"? Please precise.

10/5: Please see main comment 2.

10/8: "locally-calibrated", "on-glacier"

10/9-12: Please connect this sentence better with the rest of the discussion. Why are you discussing off-glacier temperature data here?

10/13: in converting -> to convert

10/23-24: This sentence is a bit obvious. A temperature-index model is always very sensitive to air temperature variations. Please remove or explain better this idea.

10/25-27: Check the grammar of this sentence. It is very difficult to read.

10/29-30: I am not sure if you are expressing your results correctly. Please be more precise. Based on Table 4, I would say that the average contribution is between 10% and 13% over the entire period. Individual daily values range between 3 and 34%.

10/36: high levels -> high-elevation sites

10/37: Remove "which generate more water per surface unit than the non-glaciated area".

11/1-6: This is not really a discussion of your results.

11/2: What is the elevational retreat of Universidad Glacier?

11/3-6: The idea of "peak water" is interesting. Other authors have suggested that this peak will not happen in the Andes or it already happened (Ragettli *et al.*, 2016). If you keep this paragraph, consider to extend this discussion adding more literature: (Rubio-Álvarez and McPhee, 2010) and (Cortés *et al.*, 2011) also examined streamflow trends of Chilean rivers.

11/14-15: This is not clear: "is located at a particular climatic zone which maximizes summer melting" What does it mean "to maximize summer melting"?

11/16: Check the grammar.

11/23: "estimated"

11/24: "debris-free" and "debris-covered"

11/25 "snow rich years, such as 2013-2014".

11/28: "In this study, we have investigated"

11/29: "using a distributed degree-hour melt model"

11/34: ". The ablation…"

12/7: "MacDonell et al."

12/8-9: "off-glacier air temperature measurements to the glacier boundary-layers" This sentence is not clear and you did not analyze the regional scale. For a comparison to a regional scale maybe you can use results from (Mernild *et al.*, 2015, 2016).

12/15: I am not sure if the groundwater flow is depleted in summer.

12/18: What is your source for those numbers?

12/20: "Carrasco, 2005"

12/23: "In the long term"

12/17-23: These are not conclusions from your study. They sound more like a discussion. Please move or restructure.

12/29: "thank".

12/35: add volume and page numbers

**TABLES**

Table 4: Are those max and min values daily values?

**FIGURES**

Figure 1: Please move A to the left and refer to the letters (A, B and C) in the caption (instead of upper left, etc).

Figure 2: Add letters to the panels and refer to wind speed and relative humidity in the caption.

Figure 4: Can you split this plot in several hours? similar to figure 7 in (Petersen and Pellicciotti, 2011). It would be interesting to observe the diurnal cycle of wind directions and when is the katabatic flow disrupted.

Figure 5: Can you add other reference elevations? For example, the ELA or the altitude of the AWS2.

Figure 6: Why do you cut this plot in December? Please see main comment 2.

Figure 7: "latent and sensible heat fluxes".

Is there a reason why the incoming shortwave radiation changes so sharply around January 23?

Figure 8: Another panel showing the differences between these two panels would be very informative.

How do you calculate ablation for October and November 2009 if you do not have the air temperature lapse rates for that month?

Figure 9: Why don't you show results for S1?

Why don't you show results with an uncertainty range as in Figure 6?

Figure 10: If you discarded it, don't show the HydroChile data after the earthquake.

Add the correlation coefficient you calculated in lines 9/11.

Figure 12: "and the HydroChile station"

**REFERENCES**

Ayala A, Pellicciotti F, MacDonell S, McPhee J, Vivero S, Campos C, Egli P. 2016. Modelling the hydrological response of debris-free and debris-covered glaciers to present climatic conditions in the semiarid Andes of central Chile. *Hydrological Processes* **30**: 4036–4058 DOI: 10.1002/hyp.10971

van den Broeke MR. 1997a. Momentum, heat, and moisture budgets of the katabatic wind layer over a midlatitude glacier in summer. *Journal of Applied Meteorology* **36**: 763–774 DOI: http://dx.doi.org/10.1175/1520-0450(1997)036<0763:MHAMBO>2.0.CO;2

van den Broeke MR. 1997b. Structure and diurnal variation of the atmospheric boundary layer over a mid-latitude glacier in summer. *Boundary-Layer Meteorology* **83** (2): 183–205 DOI: doi:10.1023/A:1000268825998

Cortés G, Vargas X, McPhee J. 2011. Climatic sensitivity of streamflow timing in the extratropical western Andes Cordillera. *Journal of Hydrology* **405** (1–2): 93–109 DOI: 10.1016/j.jhydrol.2011.05.013

Huss M. 2011. Present and future contribution of glacier storage change to runoff from macroscale drainage basins in Europe. *Water Resources Research* **47** (7): 1–14 DOI: 10.1029/2010WR010299

Mernild SH, Beckerman AP, Yde JC, Hanna E, Malmros JK, Wilson R, Zemp M. 2015. Mass loss and imbalance of glaciers along the Andes Cordillera to the sub-Antarctic islands. *Global and Planetary Change* **133**: 109–119 DOI: 10.1016/j.gloplacha.2015.08.009

Mernild SH, Liston GE, Hiemstra C, Wilson R. 2016. The Andes Cordillera. Part III: glacier surface mass balance and contribution to sea level rise (1979-2014). *Int. J. Climatol.* DOI: 10.1002/joc.4907

Petersen L, Pellicciotti F. 2011. Spatial and temporal variability of air temperature on a melting glacier: Atmospheric controls, extrapolation methods and their effect on melt modeling, Juncal Norte Glacier, Chile. *Journal of Geophysical Research* **116** (D23) DOI: 10.1029/2011JD015842

Ragettli S, Pellicciotti F. 2012. Calibration of a physically based, spatially distributed hydrological model in a glacierized basin: On the use of knowledge from glaciometeorological processes to constrain model parameters. *Water Resour. Res.* **48**: W03509 DOI: 10.1029/2011WR010559

Ragettli S, Immerzeel WW, Pellicciotti F. 2016. Contrasting climate change impact on river flows from high-altitude catchments in the Himalayan and Andes Mountains. *Proceedings of the National Academy of Sciences* DOI: 10.1073/pnas.1606526113

Rubio-Álvarez E, McPhee J. 2010. Patterns of spatial and temporal variability in streamflow records in south central Chile in the period 1952–2003. *Water Resour. Res.* **46** (5): 1–16 DOI: 10.1029/2009WR007982

---

## Author Comment (AC1) · 9 Jan 2017

**Author's response to Anonymous Referee #1**

We appreciate the constructive review of Anonymous Referee #1.We agree with the general comments and we introduced changes in the manuscript in order to address the reviewer's concerns. Also we clarified and corrected the manuscript considering specific comments. We think that these changes (and English and Technical corrections) benefited the manuscript, especially improving the discussion/conclusion section and providing a stronger context for the study. Here, we provide a briefpoint by point response to the general comments (enumerated) of the Referee #1:

1) **The authors must contextualize their findings with more studies about glacial hydrology, discussing more thoroughly the current status of Universidad glacier (area decrease of the last decades?) and its possible future point of peak water considering current shrinkage rates as well as local and downstream impacts of changing river runoff (in the region).**

**Author's response**

We re-wrote the Discussion section considering this comment and have added some further context to the Introduction. It is difficult to relate the results obtained from one ablation season to a long term perspective, however, considering previous research and our results we have discussed the future runoff and the relatedimpacts on water availability in the region. To address this point we added more literature discussion regarding future runoff trend in the region.

2) **It is not clear, if the results should be seen as a first short snapshot (only 5-6 months measurements) at the beginning of an anomalous period of drought (2010-2015) or if they can be brought into a wider context (ideally with longer in-situ data)**

**Author's response**

We have clarified this point (also made by two specific comments of this reviewer). We focused on one ablation season due to the availability of data (after March 2010 no more data/observations were obtained from the glacier). Hence, we prefer touse the results as representative of certain synoptic/weather conditions. The analyzed period coincided with the beginning of the "mega-drought" that affected central Chile in the last 7 years so we assume that the results are representative of a dry period.

3) **While relative glacier melt fraction to river runoff might be high particularly in dry periods and the upper Tinguiririca catchment, relative contribution is expected to decrease with increasing distance from headwaters, i. e. for the**

**low-lying coastal cities and water users. The mention of (the insignificance of) groundwater flows, probably difficult to estimate without direct measurements / tracing methods, should be revised as many different hydrological models have not been capable to adequately represent groundwater flows. Some studies of the last years suggest that they represent an important driver (e. g. Baraer et al., 2014 for the outer tropical Andes of Peru).**

**Author's response**

We agree with this comment, therefore we changed the text adding, discussing related literature as suggested.

4) **The manuscript contains multiple tables and figures, most of them helpful for further comprehension, others less substantial. In order to reduce total paper volume, I would skip e.g. Table 1 and Figure 4. However, all anomalies / data gaps in the plots should be briefly indicated and explained in the text or subtitles.**

**Author's response**

We remove Table 1 as the referee#1 suggested, however we keptFig. 4 since  Referee #2 recommended to improved and not skipped.This implies hourly wind rose to see if katabatic flow is interrupted during the evening. We explain data gap and anomalies in the data, as specified in Reviewer's specific comments.

**Response to specific comments:**

All the specific comments and technical correction were addressed in the revised manuscript.

**1 / 10-11: is that true that glacier melt represents more than the half of total streamflow contribution in lowlands during dry years in Chile? I would rather expect a reduction of relative contribution with increasing distance from the glacier and headwaters converting glacier streamflow to an important but not the main contributor in the lowlands.**

We agree that the manuscript was not clear on this point. Considering that the comparison of glacier melt with observed runoff in this research, as withmany previous studies,was made with data from stations located in the upper part of the mainvalleys, it is difficult to make an assessment of glacier contribution to runoffin the central valley of Chile or on the coast. We clarify this point, in the Abstract and in the Discussion section 4.2.

**2 / 38 – 3 / 1: what about glacier area and (estimated) volume changes and current retreat rates of Universidad glacier and/or in the region?**

We didn't estimate the area/volume changes for Universidad glacier. We used available literature (Le Quesne et al., 2009; Wilson et al., 2016) regarding these changes in the last years. However we added more literature to contextualize the glacier reduction in the region.

**3 / 7-8: you identify the year 2009/2010 as (just) the beginning of a longer dry season (2010-2015) but it is unclear why you did not incorporate a longer period of measurements into you study** and**3 / 18-19: again, you do not explain why your study only covers six months of data measurements**

We clarify this point. See General comment 2).

**3 / 33-34: how did you discriminate snow from ice with the NDSI? Thresholds and techniques should be mentioned 4 / 3: clarify which images were selected with a cloud cover threshold: Landsat 5 TM?**

We have stated more clearly that only MODIS products were used for snowline elevation identification and snow/ice discrimination. The mention of Landsat relates to earlier work not done by us to improve retrieval of sub-pixel snow cover information, and possibly this has been a source of confusion. We have usedthe MOD10A1 product since provides a better differentiation of the ice surface of Universidad glacier (which is dirty due the presence of ogives, debris and impurities), from and the fresh snow areas. However, MOD10A1 product gives the fractional snow cover for each pixel in the range 0 to 100, and to assure a correct snowline altitude we assumed the presence of snow in the pixel with a fractional value of 100. Despite this we expect some uncertainties in the snowline altitude as Fig.5 shows for the end of the ablation where high variability exists.

**4 / 14-15: the explanation of how to convert hourly to daily format is very basic and can be neglected**

OK. Deleted

**10 / 29-31: again, be careful that you distinguish upstream from downstream (lowland) glacier streamflow contribution, the latter possibly less significant; what about flow contribution in australwinter? Although you have only worked in the ablation period, it would be good that the reader gets a general idea of glacier streamflow contribution changes during a whole hydrological year.**

We added more literature regarding streamflow contribution during a complete hydrological year.

**11 / 3-6: the point of (future) peak water is not sufficiently investigated in many mountainous regions worldwide but an increasingly important research question, particularly for future water management, can you examine this question about the possible peak water of Central-Andean glaciers in Chile a bit more? More literature?**

We added more literature discussing this point. (See General comment 1).

**11 / 13-14: is it true that melt rates are generally reduced further north (until where?) of Universidad glacier? Sublimation process are strongest with a pronounced water vapor gradient which is true for the dry season of e. g. the outer tropics (Peru/Bolivia) but not for glaciers in the inner tropics.**

We were referring to high altitude glaciers in the north of Chile and in the outer tropics from Peru and Bolivia. We clarified this point in the new version of the manuscript.

**11 / 37 – 12 / 1: is Universidad glacier really such a particular glacier with highest melt rates in Chile? cite comparing literature**

We compared our results with other studies in section 4.3. However, as referee 2 suggests we provided more data from previous studies to establish a meaningful comparison.

**12 / 2-3: this affirmation is obsolete as it represents a typical mechanism of glacier energy budget and mass balance**

OK. Deleted

**12 / 15: does groundwater flow really becomedepleted? any studies (e. g. tracers: Rodriguez et al., 2014)? in other parts ofthe Andes (where reduced ablation also takes place during the winter season) groundwaterhas been identified to be a strong contributor and generally underestimated inmany studies**

We reviewed this point and changed the sentence considering that groundwater could be a major contributor. (See also General Comment 3).

**12 / 24: the last argument should be more developed. The region isimportant for multiple water users. As an example, just some kilometers downstream,the hydropower plants La Higuera / La Confluencia are situated and possibly stronglyaffected by annual/seasonal changes in river runoff**

We added this information about the water used by hydropower as well as for agriculture activities in this particular basin.

**16 / Table 1: this table does not contribute substantially to the study comprehension, therefore I would take it out**

OK. Deleted

**18 / Table 3: indicate period in the title "(2009-2010)"**

Added

**20 / Figure 1: upper left: the three gauges are not clearly identifiable; the map text "CECs HydroChile" confuses; also, the abbreviations "CECs" and "DGA" in the legend are not proper; text of the figure: add "(orange outline)" after "Universidad glacier"**

OK. Changed and added accordingly.

**21 / Figures 2-13: indicate altitude (m asl) for ALL station data**

OK. Added

**23 / Figure 4: in order to reduce paper volume, I would skip this graph as it does not substantially contribute for a further process comprehension**

We have preferred to kept this Figure with some changes suggested by referee 2.

**25/ Figure 6: eliminate "[dd-mmm-yyyy]" at x-axis legend; you also do not use this definition in Figure 7**

OK. Deleted

**26 / Figure 7: indicate gaps which are present between November 21-22**

OK. We have added and explained each data gap

**Figure 12: no runoff measurements from March on? explain this data gap**

We explained that period of observations (AWS, PS) is until end of March. See General Comment 2).

**Technical corrections**

**1 / 1-3: with 28 words, the title is too long and complicated. Amore concise title would be: "Glacier melt contribution to river runoff at Universidadglacier, central Andes of Chile"**

We changed the title as suggested

**1 / 11: eliminate "the" before "glacier melt"**

Done

**1 / 13: insert"within the" before "central Andes of Chile" 1 / 19: replace "altitude part" by "ablationarea"**

Done

**1 / 28: insert "a" before "crucial resource"**

Done

**2 / 21: change order "Mediterraneanclimate type"**

Done

**2 / 26: use directly the previously introduced abbreviation "AWS"**

Done

**2 / 32:correct "altitudinal range"**

Done

**2 / 33: improve "which converge at an altitude"**

Done

**2 / 35-37:change order considering a clockwise aspect of glaciers (north to the west)**

Done

**3 / 2-3:"fastest period" does not exist, improve**

Changed

**3 / 9-10: three times the word "measurements",Replace**

Replaced: "Data collected include meteorological observations at two AWS, surface lowering monitoring from ablation stakes and a sonic ranger (Fig. 1), satellite-derived snow cover distribution and discharge measurements in the proglacial stream"

**3 / 14-15: not a full phrase, a verb is missing!**

We have added the verb

**3 / 16: correct "net all-waveradiation"**

Done

**3 / 32: insert "spatial" before "resolution" (there are also other types of resolutions)**

Done

**3 / 33: better specify "Landsat 5 TM (30 m spatial resolution)"**

Done

**4 / 1-2: eliminatethe long parenthesis "(Advanced Spaceborne. . . Version 2)"**

Eliminated

**4 / 30-31: improve phrase:it is not "melt overestimation" which is dominated by melt from the ablation zone; insteadof "however" you could use "as it"**

Done

**5 / 5: "and the afternoon maximum" could bethe beginning of a new phrase and needs a verb**

Changed

**5 / 14: include "shortwave" before "radiation"**

Done

**5 / 21: insert "to be" before "a constant"**

Done

**5 / 22: add "a" before "function"**

Done

**8 / 2:eliminate "(100% relative humidity" – very basic**

Done

**8 / 4: correct "was covered"**

Done

**8 / 21-22:improve phrase, you could separate it into two phrases from "fluxes calculated by" oinserting a new verb**

We rather prefer to kept this phrase adding "turbulent" after the first comma.

**10 / 3: eliminate "a" before "suitable"**

Done

**10 / 4: use also the word"correlation" instead of only "agreement"**

Done

**10 / 8: improve, e. g. "an hourly calibratedlapse rate at the glacier"**

Done

**10 / 19: maintain the same terms, here "Universidad glacier"**

Done

**10 / 37: replace "that" by "than"**

Done

**11 / 14: add "cover" after "cloud"**

Done

**11 / 20: eliminate"the" before "each"**

Done

**11 / 24: add "the" before "central Andes"**

Done

**11 / 25: correct "dependson"; insert "as" before "2013"**

Done

**11 / 26: correct phrase "while in dry years"**

Done

**11 / 28: betterwrite "climatic conditions" instead of "meteorology"**

Done

**11 / 29: insert "model" after "melt"**

Done

**11 / 34: a final point is missing before "The ablation"**

Added

**11 / 37: improve phrase avoidingthe semicolon with e.g. "and are thus greater"**

Changed

**12 / 2: "latitudinal" instead of "latitude"**

Changed

**12 / 3: "persistent" instead of "persistence"**

Changed

**12 / 11: add "km" after "1.7"**

Added

**12 / 12: aspace is missing before "The total"**

Added

**12 / 13: improve phrase: "Universidad glacier onlyrepresents 36%"**

Changed

**12 / 14: add the year "2010" after "March"**

Added

**12 / 20: insert "the" before"zero-degree"**

Added

---

## Author Comment (AC2) · 9 Jan 2017

**Author's response to Anonymous Referee #2**

We would like to thank anonymous Referee #2 for his/her constructive and useful comments that have significantly improved the manuscript. We agree that discussion and conclusion needed to be improved (as Referee#1 also suggests), so have changed the text accordingly.Here, we provide a brief point by point response to the general comments (enumerated) of the Referee #2:

1) **Specify if the glacier contribution to runoff is coming from the snow or the bare ice.**

**Author's response**

This is a good suggestion. As the referee indicates we can separate between melt from snow over the glacier and bare ice. We added this information in the manuscript.

2) **Validation of the degree-hour method in Figure 6**

**Author's response**

We have extended the comparison between EB and temperature-index model until the end of January 2010improving the validation of the degree-hour method. After that date, and as the referee noted in the minor comments for Fig.7, the radiation (shortwave and net) was reduced sharply. So we did not use data of February and March.

3) **Comparison to other studies (section 4.3)**

**Author's response**

We added the data that the referee suggested.

4) **DHF for snow**

**Author's response**

We used all time steps (24 h * 52 d) to estimate the DHF for snow, since the hours with negative temperatures were set to 0°C to calculate the mean temperature. With this procedure we obtained a DHF of 0.1188 mm we h$^{-1}$ °C$^{-1}$.However, the percentage of hours with negative temperatures during the period were close to 25%. Using only the positive time steps, the mean temperature was4.6°Cwhich gives a DHF for snow of 0.1192 mm we h$^{-1}$ °C$^{-1}$. The impact of these changes on the DHF value is therefore negligible. Anyway, we changed the explanation of this procedure in the text.

**Response to minor comments**

In general we agree with all suggested text corrections and minor comments. Here,we response directly every minor comment:

**Please correct systematically throughout the paper the use of capital letters for glaciers and rivers. Universidad glacier -> Universidad Glacier, Tinguiririca river ->Tinguiririca River.**

Done

**Check the use of present and past tenses. You frequently change from one tense to another, especially in the methods section.**

Checked

**1/11: "during late summer and autumn"**

Changed

**1/12: "To address these shortcomings", why in plural? You only describe the fact that few studies are available.**

We changed to singular

**1/15: "to compare"**

Changed

**1/28: "water is a crucial resource"**

Changed

**1/28-29: Could you add more references apart from Masiokas et al. 2006 to sustain this sentence? The study of Masiokas et al. 2006 is about snowpack variations and not water uses.**

We discussed and added more references regarding water uses (e.g. Meza et al., 2013).

**1/30-31: Please reword: "In this region, winter precipitation is driven by the interactions…., and summer runoff (or summer water supply) by the storage and release from glaciers and the seasonal snow cover".**

Done

**1/32: Maybe replace "water supply" by "runoff generation".**

Done

**1/35: I think you need to replace "altitude" by "elevation" if you refer to terrain. How do you calculate the 4000 m asl? Is it the average of the peaks? Do you have a reference?**

We replaced "altitude" by "elevation" and added a reference indicating the mean altitude of the Andes in the region.

**2/2: "warm temperatures"**

**Consider to change "trigger" by "produce" or "cause"**

Done, we changed "trigger" by "cause"

**2/3: "rivers in the Andean basins of central Chile are mainly driven by the melting of the seasonal snowpack."**

**Please specify that you refer to the highest river sections. The annual regime is driven by winter precipitation in the lower sections.**

We explained "high" was related to  Central Chile Andes basins

**2/3-4: The expression "is related to the existence" sounds awkward.**

We changed by "is related to the presence"

**2/6-7: "For example, Peña and Nazarala (1987) estimated that the contribution of glacier melt to the Maipo River basin in summer 1981/82 was maximum in February and represented 34% of total discharge".**

**Please provide the elevation of the outlet and the percentage of the glacierized area of the catchment analyzed by Peña and Nazarala (1987). Also if by "glacier melt" they include the seasonal snow over the glacier.**

We provided the information of the outlet. On the other side it is not clear in the work of Peña and Nazarala (1987) if glacier melt include snow over the glacier. However the 34% contribution corresponds to February of a dry year. The same authors indicate that "snow remaining at the end of a dry year is negligible". This makes suppose that the indicated percentage is referred only to ice.

**2/10-12: Please provide one or two sentences with the main conclusions of Pellicciotti et al. (2008). Otherwise this reference is not very meaningful.**

We added "…zone, showing that the ablation process is dominated by incoming shortwave radiation."

**2/12-14: Please provide the elevation of the outlet and the percentage of the glacierized area of the catchment analyzed by Ragettli and Pellicciotti (2012). Also if by "glacier melt" they include the seasonal snow over the glacier.**

We added this information

**2/14-16: Please check this sentence. "Results are available only for one basin" sounds strange.**

We changed to "Despite these advances, results are limited to one basin and cannot necessarily be extrapolated"

**2/17: "or on the impact of…"**

Done

**2/18-19: can you be more specific? What do you mean by "melt patterns"? Temporal, spatial?**

Both, we explained this in the text

**2/20: deficiencies -> knowledge gaps, issues**

Done

**2/33: the words "which convergent" are not clear.**

Re-written; "which converge at an altitude of ~2900 m asl"

**From where did you obtained the ELA? References?**

We re-estimated the ELA. See below Minor comment 8/3-6

**2/33-34: Below this ->Below this elevation, below the ELA**

Changed: "below this elevation"

**3/10: "After the analysis of energy fluxes at the location of the lower AWS, a temperature-index model was calibrated and applied at the glacier scale. Resulting melt amounts were used to estimate total glacier discharge, which is compared with downstream discharge records."**

Changed as suggested

**3/21: Please choose another title. Snow density is not an ablation measurement.**

Re-written: "Ablation measurements: stakes and sonic ranger"

**3/22: re-measured -> read**

Changed

**3/26: melt -> of surface ablation. The ablation stake also includes sublimation.**

Changed

**3/27: "(Table 1). The sensor recorded surface …".**

Changed

**3/31-32; Please provide more details about the regression between Modis and Landsat products.At least the basic principles.**

We added more literature

**3/35: elevational distribution of snow cover -> snow line**

Changed

**4/1: What is the acquisition date of the DEM? Is it similar to that of the study period?**

Unfortunately ASTER GDEM did not give time acquisition. We added this point as one uncertainty in the melt estimation.

**4/3: "Images were used…" ,Modis images?**

We changed by "MODIS product were used…"

**4/3-4: Remove "For modelling purposes".**

Removed

**4/8: remove the second "applied".**

Removed

**4/10-11: ", which we refer to as the degree-day factor, …".**

Changed

**4/11-12: stake 1 melt measurements -> stake 1 ablation measurements.**

Changed

**4/11-13: This sentence is not clear. Please reword, perhaps you should split it in two.**

Re-written; "We use stake 1 ablation measurements (Table 2) and the mean positive air temperature (3.5 °C)  at the AWS1 to estimate a DHF for snow. Dividing the ablation value by the mean of positive air temperature (Braithwaite et al.,1998), we obtained a DHF for snow of 0.12 mm w.e. h$^{-1}$ °C$^{-1}$."

**4/12: "negative temperatures are set to 0°C" This sounds very strange. Say instead that you set to zero all melt occurring at time steps when the air temperature is below the temperature threshold.**

Deleted

**4/13: Please place the value 3.5°C in another part of the sentence.**

Done

**4/13 "With these values" What values?**

Changed

**4/13: "Following the procedure of Braithwaite et al. (1998)" What procedure? Please briefly explain it. Do you divide the total ablation by the total number of hours or only by those with positive air temperatures?**

For the total number. We have clarified  this point. See general comment 4).

**4/14: Why would you multiply by 24? I would think that melt only occurs during daytime (maybe 14 or 16 hours per day).**

See general comment 4).

**4/16: Please add at the beginning of the sentence a short explanation of why you cannot use the same procedure as for snow: "As we do not have ablation stake measurements in the period when the ice surface is exposed, we use a range of published….".**

Added

**4/21: Can you use only one symbol? Either Df or DHF.**

We prefer to keep both symbols. The first one is the mathematical symbol for the equation as the editor suggests. If we use DHF in the equation, this means that D times H times F (which is not correct). For the text we prefered to maintain DHF to reduce the extension of the text.

**5/3: Since these are negative values, maybe write "with a minima in magnitude".**

Changed

**5/6: It should be "entrainment of warm air from the upper atmospheric layers". Please see the articles from van den Broeke (1997a, 1997b) in Pasterze Glacier for a more theoretical perspective. Insolated bare rock surfaces can also locally increase near-surface air temperature, but I don't think that "entrainment" is the right term.**

We changed by "advected"

**5/6-7: Could you please check if wind directions reveal up-winds from the proglacial valley? Petersen and Pellicciotti (2012) observed this feature in Juncal Norte Glacier.**

We check this. During the afternoon, data from AWS1 show up-winds. However katabatic winds still are the prevalent during all day. We showed this in a new Fig. 4

**5/12: "was determined following Oerlemens (2010)". Remove also the parenthesis.**

Changed

**5/17: Why do you need the reference of Oerlemans and Klok to neglect the heat from the rain? Or is the reference wrongly placed?**

We used this reference as an example, since they neglected heat from rain in their calculations. We added e.g. in the reference to clarify.

**5/19: "The sensible heat fluxes were calculated…"**

Changed

**5/22-23: Do you assume the same value of z0 for snow and ice?**

No. We assumed to be 0.001 m for melting snow and 0.01 m for ice on mid latitude glaciers (Brock et al., 2006). We added in the manuscript.

**6/3-5: I guess this is ok, but you are assuming the surface temperature as 0°C for the sensible heat fluxes, so, to be consistent, everything should be evaporation.**

That's right we clarified this point in the manuscript

**6/13: You missed the evaporation rate.**

Added.

**6/26: Do not mention what you did not do, delete "There were no direct measurements…"**

Changed

**6/28: Add a space before "Water level…"**

Added

**7/30: replace "almost always" and "more frequently" by a percentage of time.**

Replaced

**8/3-6: Can you say something about the ELA with this procedure? If we use the elevation of the snowline at the end of the ablation season as an indicator of the ELA that year, we would get a number much higher than the value of 2900 m asl (mentioned in line 2/33).**

Effectively the ELA was wrong (too low). We changed the value at 2/33 and explained the estimation procedure. Also we discussed the value obtained from the ASTER image (Fig1) and the values obtained using MOD10A1

**8/14: Please use the same number of significant digits for the DHFs (in lines 4/18 you use only two).**

Ok, we used two all along the text.

**8/18-19: Please see main comment number 2.**

ok

**8/32-33: Move to methods.**

Done

**9/3: October -> November**

Changed

**9/15: "purposes".**

Done

**9/19-20: Do you have data before November 24? Why don't you start the comparison on October 1?**

Yes we have AWS data before this date, however the pressure sensor only have data from this date and on

**9/23: "…contributions from glaciers…"**

Changed

**9/23: Please mention these lakes in the catchment description.**

Added

**9/29-31: Please explain this sentence better:**

**"At the beginning of the common period": What period do you mean exactly?**

We added the explanation about the period. **"in the basin": what basin? The largest one?**

We referred to the entire high Tinguiririca basin (added)

**Why is the high daily variability associated with the control of air temperature over snowmelt?**

The interdaily variability of the air temperature is similar to the interdaily variability of the runoff in the DGA station. We added air temperature as reference in Fig. 13.

**10/3: "is suitable"**

Changed

**10/4: "high melt regime" is not a very precise term. Do you mean something like "large retreat during last years"? Please precise.**

We deleted "high melt regime" to avoid confusion

**10/5: Please see main comment 2.**

Ok

**10/8: "locally-calibrated", "on-glacier"**

Changed

**10/9-12: Please connect this sentence better with the rest of the discussion. Why are you discussing off-glacier temperature data here?**

**10/13: in converting -> to convert**

Changed

**10/23-24: This sentence is a bit obvious. A temperature-index model is always very sensitive to air temperature variations. Please remove or explain better this idea.**

Removed

**10/25-27: Check the grammar of this sentence. It is very difficult to read.**

Changed

**10/29-30: I am not sure if you are expressing your results correctly. Please be more precise. Based on Table 4, I would say that the average contribution is between 10% and 13% over the entire period. Individual daily values range between 3 and 34%.**

Yes, we changed 10% by 3%

**10/36: high levels -> high-elevation sites**

Changed

**10/37: Remove "which generate more water per surface unit than the non-glaciated area".**

Removed

**11/1-6: This is not really a discussion of your results.**

We used Carrasco et al. (2005) and LeQuesne et al. (2009) as general precedent relating frontal retreat and temperature trends in the glacier. We agree that it is not directly a discussion of our results, instead is a discussion of the implications of the context of glacier retreat and future melt.

**11/2: What is the elevational retreat of Universidad Glacier?**

Elevation retreat is close to 70 m between 1955 and 2007 (LeQuesne et al., 2009).

**11/3-6: The idea of "peak water" is interesting. Other authors have suggested that this peak will not happen in the Andes or it already happened (Ragettli*et al.*, 2016). If you keep this paragraph, consider to extend this discussion adding more literature: (Rubio-Álvarez and McPhee, 2010) and (Cortés *et al.*, 2011) also examined streamflow trends of Chilean rivers.**

We added the suggested references to the discussion.

**11/14-15: This is not clear: "is located at a particular climatic zone which maximizes summer melting" What does it mean "to maximize summer melting"?**

**11/16: Check the grammar.**

Ok

**11/23: "estimated"**

Changed

**11/24: "debris-free" and "debris-covered"**

Changed

**11/25 "snow rich years, such as 2013-2014".**

Changed

**11/28: "In this study, we have investigated"**

Changed

**11/29: "using a distributed degree-hour melt model"**

Changed

**11/34: ". The ablation…"**

Changed

**12/7: "MacDonell et al."**

Changed

**12/8-9: "off-glacier air temperature measurements to the glacier boundary-layers" This sentence is not clear and you did not analyze the regional scale. For a comparison to a regional scale maybe you can use results from (Mernild*et al.*, 2015, 2016).**

We clarified and addressed this point.

**12/15: I am not sure if the groundwater flow is depleted in summer.**

As referee 1 suggested we introduced changes in this topic, adding more literature discussion.

**12/18: What is your source for those numbers?**

We added a reference for SST

**12/20: "Carrasco, 2005"**

Changed

**12/23: "In the long term"**

Changed

**12/17-23: These are not conclusions from your study. They sound more like a discussion. Please move or restructure.**

Moved to discussion

**12/29: "thank".**

Changed

**12/35: add volume and page numbers**

Added

**TABLES**

**Table 4: Are those max and min values daily values?**

Yes. We added this in the explanation

**FIGURES**

**Figure 1: Please move A to the left and refer to the letters (A, B and C) in the caption (instead of upper left, etc).**

Changed

**Figure 2: Add letters to the panels and refer to wind speed and relative humidity in the caption.**

Added

**Figure 4: Can you split this plot in several hours? similar to figure 7 in (Petersen and Pellicciotti, 2011). It would be interesting to observe the diurnal cycle of wind directions and when is the katabatic flow disrupted.**

Changed

**Figure 5: Can you add other reference elevations? For example, the ELA or the altitude of the AWS2.**

Added

**Figure 6: Why do you cut this plot in December? Please see main comment 2.**

We changed the extension. See General comment 2).

**Figure 7: "latent and sensible heat fluxes".**

Changed

**Is there a reason why the incoming shortwave radiation changes so sharply around January 23?**

There is not a clear reason so we preferred discarding this data. See General comment 2).

**Figure 8: Another panel showing the differences between these two panels would be very informative.**

We added a new panel with these differences. We also indicated how much corresponded to ice melt and how much corresponded to snow melt on the glacier.

**How do you calculate ablation for October and November 2009 if you do not have the air temperature lapse rates for that month?**

We assumed the same lapse rate observed in the common period. We clarified this in the manuscript.

**Figure 9: Why don't you show results for S1?**

Result from S1 are showed in Fig. 6

**Why don't you show results with an uncertainty range as in Figure 6?**

We only plotted results from one DHF (0.38 mm w.e. h-1 °C-1). However we added both result to show the uncertainty range.

**Figure 10: If you discarded it, don't show the HydroChile data after the earthquake.**

Done

**Add the correlation coefficient you calculated in lines 9/11.**

Added

**Figure 12: "and the HydroChile station"**

Changed

---

## Referee Report (RR1)

**Review of manuscript (major revision)**

Journal: HESS
Title: **Assessing glacier melt contribution to river runoff at Universidad glacier, central Andes of Chile**
Author(s): Claudio Bravo et al.
MS No.: hess-2016-503
MS Type: Research article

**General comments**

The authors have appropriately addressed most specific and technical corrections suggested before. My impression is that the manuscript has substantially gained in both a precise description of methods and results as well as a more thorough discussion putting modelling results into a broader context e.g. for water users in the region and future research.

Unfortunately, there is confusion about the percentages of glacier contribution and the corresponding period. Some of the new text inserts again do not meet publication criteria due to typing and grammar errors (see e.g. p.11!). Please take this serious, I mentioned these evident deficiencies through the whole manuscript before. Furthermore, some last doubts/imprecisions should be addressed.

In summary, I recommend minor revision correcting these last issues detailed below.

**Specific comments**

1 / 18: does glacier contribution account for 10-13% for the whole period (November 2009 to March 2010) or even 3-34%? do you really differentiate between November to March and December to March? quite confusing! Table 3 does not include November - I guess you have not revised and adapted all sections, do so now!: 1 / 18;  10 / 03;  10 / 9-11;  11 / 06;  13 / 13

3 / 21: the 'mega-drought' conditions in Central Chile are still prevailing, right? considering that we are already in 2017 and you want to emphasize this present effect of drought you could extend "2015-2016" (and, if available, include some latest bibliography) – also in the other sections

10 / 21-23: the good agreement you mention does not result from data availability itself, improve this part

11 / 17-18: is the "positive trend" of 0.3 m³/s/y significant or not? if not, you cannot refer to a "positive trend"

11 / 31: start a new sentence "However, the possibility…" and sharpen your argumentation as  a) I understood that drought conditions are still going on and  b) this sentence calls for a consequence, e. g. "… in the future and therefore more research is needed in order to address these issues."

12 / 07-10: the periods are both three months, from January to March for 2004 and 2005 with corresponding melt values of 4950 mm w.e. (2004) and 3960 mm w.e. (2005)? a little bit confusing

12 / 29-30: both studies you are mentioning in the context of the importance of groundwater flows were performed in climatically different regions, the outer tropics in Peru and central Chile. Baraer et al. 2014 refer particularly to the dry season (of each year) - please clarify

13 / 32: although it is very likely, please cite your statement "increasing demand for water in the region"

**Technical corrections**

1 / 19: better reduce "during November 2009 and March 2010."

2 / 02: Delete (Garreaud, 2013), already cited in line 3

2 / 38: correct "between November and January"

3 / 04: "depending on" the aspect

3 / 18-19: delete "(after March 2010 no more data/observations were obtained from the glacier)", no additional information

3 / 26-27: three times "installed", avoid redundancies

4 / 17-18: "dirty" is a very colloquial and "impurities" a very broad term, better write "partially covered by debris and aerosols"

4 / 30: add "up" after "summed"

4 / 31-32: be consistent with the use of units "between 7 mm w.e. °C$^{-1}$ d$^{-1}$ and 9 mm w.e. °C$^{-1}$ d$^{-1}$" or just "from 7-9 mm w.e. °C$^{-1}$ d$^{-1}$"

5 / 16: again "between … and …"

5 / 28: use past instead of present "We restricted"

5 / 29: be consistent with other date formats in your manuscript "29 January"

7 / 14: better write "from a catchment with a total area of 86 km² which is partially covered by…"

7 / 24: use singular "The geometry... was measured"

8 / 16: again "between … and …"

9 / 13-14: avoid redundancies "some melting, with values around…"

9 / 22: use "16:00 h… 10:00 h" or "4:00 pm … 10:00 am", be consistent with the same format used in 8 / 16-17 and other sections

10 / 08: correct "Tinguiririca basin" and add "to" after "due"

11 / 14: divide "Tinguiririca basin"

11 / 19: again "between 1975 and 2001"

11 / 21-22: better write "positive discharge trends for several rivers in central Chile"

11 / 23: correct "Tinguiririca"

11 / 24: "indicating that"

11 / 24-25: be careful, a typical grammar issue in your manuscript "tends to occur"

11 / 26: correct "it is uncertain" and write "whether" (avoidable with a normal grammar check!)

11 / 28: correct "is yet to occur"

11 / 30: use plural "Estimations of"

11 / 37: use the English form for "Peru"

11 / 39: eliminate "due to increasing latitude" and use plural "reduce"

12 / 01: use plural "Local factors"

13 / 10: add "asl" after "~3800 m"

13 / 28: eliminate first "forestry" and "irrigation", both are redundant

13 / 30: not clear "medium to long term trends" – you refer to future discharge trends, clarify

13 / 31: more precise is "glacier contribution to river discharge"

---

## Referee Report (RR2)

Review for Hydrology and Earth System Sciences

**Title: Assessing glacier melt contribution to river runoff at Universidad glacier, central Andes of Chile**

Authors:  Bravo, Loriaux, Rivera, Brock

**PAPER SUMMARY AND RECOMMENDATION**

Bravo et al. estimate the runoff contribution of Universidad Glacier (~34°, central Chile) to the upper Tinguiririca River catchment (outlet at 560 m asl) during the austral summer 2009-10. The authors use a set of meteorological, glaciological and hydrological measurements to run a point-scale energy balance model and to calibrate a degree-hour model, which is later used to calculate melt over the entire glacier extent. The authors find that glacier melt rates are extremely high (>10 m w.e.) at the glacier terminus and that the runoff contribution of the glacier represents a 10-13% of the summer runoff of the upper Tinguiririca River catchment. This contribution reaches almost 20% during late summer (March) with daily peaks of 34%. They also conclude that a temperature-index model provides good estimations due to the availability of on-glacier data, the large observed melt rates and the use of diurnally varying lapse rates.

The topic addressed by this study is appropriate for the scope of HESS. Despite the key hydrological contribution of Andean glaciers to the semiarid catchments of central Chile and Argentina, on-glacier meteorological and hydrological data are still rare and few studies have explicitly simulated melt and runoff from glaciers in this region.

The authors have properly answered to the comments of the reviewers regarding the methodology, which is now very clearly presented. Figures and Tables are of good quality. I do not think that much more work is needed to correct or produce new results. However, I think that the Discussion and Conclusions sections need to be substantially improved. These sections present arguments that are not clearly justified (please see major comments) and there are still many necessary corrections of style and grammar (see technical corrections) before the article is acceptable for publication.

**MAJOR COMMENTS**

*1. Conclusions*

Some conclusions are confusing and not all of them are properly supported by the obtained results. The authors might consider restructuring the conclusions based on to the proposed objectives. I copy here the conclusions:

a. "Good agreement was found between melt estimated from degree-hour and energy balance models, and ablation stake and sonic ranger records at the lower weather station site in the ablation zone, supporting the application of a simple temperature-index method of calculating total glacier melt at this location. The degree-hour model was distributed at the glacier wide scale accounting for hourly variations in the local temperature lapse rate, which tended to be shallower during the daytime, when most melt occurs."

This conclusion is very confusing, as it mixes results ("Good agreement was found…") and methods ("The degree-hour model was distributed…"). In my opinion, the most relevant conclusion in this paragraph is that a degree-hour model provides a good simulation of surface lowering at the glacier tongue. This is probably because surface temperature is constantly close to 0°C and negative latent heat fluxes are negligible.

b. "The ablation regime is dominated by incoming shortwave radiation, with highest melt rates occurring during December to February, and is also characterized by high air temperature which is almost continuously positive on the lower ablation zone between November and March. These climatic conditions result in very high melt totals, which exceed 10 m w. e. melt on the lower tongue and are thus greater than melt values reported for other glaciers in central Chile. This is attributed to the relative insignificance of sublimation to total ablation, and the high insolation due to low cloud cover and latitudinal location, combined with predominantly positive air temperature. Melt totals were much lower in the accumulation area due to lower temperatures and persistent snow cover above about ~3800 m."

As incoming longwave radiation was not measured, I'm not sure if the authors can state that incoming shortwave controls the ablation regime. Or what do they mean exactly? That melt correlates to shortwave radiation or that incoming shortwave is the main energy input? Please be more explicit. In fact, at the end of January, incoming shortwave clearly decreases, but the energy available for melt keeps constant (or even increases) (Figure 7). Furthermore, if the regime is dominated by shortwave radiation a good explanation must be provided for the successfully use of a temperature-index model. Would an enhanced-temperature index model (that includes shortwave radiation) improve simulations?

The comparison with other glaciers does not support the hypothesis that melt rates on Universidad Glacier are larger than those on other glaciers of the central region. Only one glacier in the central region is used for comparison (Juncal Norte Glacier), but the simulated period was different (I think that Pelliccotti et al. 2014 used data described in Ragettli and Pelliccotti 2012: only the period December 2008 to February 2009). Furthermore, only one season at each glacier is not enough to provide a meaningful comparison. The hypothesis of large melt rates at Universidad Glacier is interesting, but should be better justified. If this is the case, is Universidad Glacier retreating faster than other glaciers? Has this been observed? Is there any other evidence that supports this hypothesis? Surface sublimation on Juncal Norte is also negligible on the glacier tongue (Pelliccotti et al. 2008).

"Melt totals were much lower in the accumulation area…" This is obvious. Please delete or explain further.

c. "During the late ablation season, in February and March 15 2010, when other runoff sources such as snowmelt become depleted, the daily contribution of Universidad glacier to total runoff in the Tinguiririca reached as high as 34%."

As the authors state that there is a 1 or 2 days of lag between the simulated runoff from Universidad Glacier and streamflow measured at the DGA station (lines 10/15), daily values should not be used to estimate the percentage of glacier contribution. Calculations should be made only at the monthly scale, i.e. the 3rd and 4th column of Table 3 should be deleted.

2. Discussion
a. "Forcing temperature-index models with off-glacier data is problematic due to the depression of near-surface air temperature within the glacier boundary layer (Shea and Moore, 2010) under positive ambient temperature conditions"

Actually, many researchers argue that on-glacier data is affected by the glacier surface temperature and that temperature-index models should be forced only with off-glacier temperature data, because it is a better indicator of ambient conditions. As the authors did not evaluate the degree-hour model using off-glacier data, they cannot state that on-glacier data works better than off-glacier on this glacier.

b. Others comments

The discussion of the uncertainties misses the uncertainty in degree-day factors, which is actually very large. Figure 12 shows that the uncertainty in cumulative runoff at the end of the season is about 7±1 m^3 x 10^7, i.e. about 15%. As the streamflow at the DGA station is very large, this uncertainty is translated as a small percentage of total streamflow (10-13%). Hock (2003) provides a table with a large range of values, how did the authors choose the range between 7 and 9 mm d-1? Did they do some previous tests or used another reference? Another important missing source of uncertainty is the calculation of the snow line from MODIS.

The authors should add a discussion about the shortcomings of using a temperature-index model in relation to a more physically-based model. This is one of the aims proposed by the authors (number 2) Do you think that the degree-hour model could be missing something important? Maybe shading effects? Radiation fluxes? Is the model appropriate to simulate long-term mass balance or should only be used to simulate ablation at the glacier tongue?

Please include in the discussion the influence of debris-covered areas on the glacier tongue. Are the large observed melt rates substantially reduced by the thermal insulation of debris? Maybe those values above 11 m are not realistic?

**MINOR COMMENTS**

The Introduction could benefit from a better structure in which climate, hydrology and economic characteristics are separately described.

Please check the use of terms "runoff" and "streamflow". I think that they are not the same: Runoff is the portion of precipitation or melt that does not infiltrate or evaporate. Streamflow is the runoff of surface water through a channel.

2/22: Why is 40°S important?

2/27: "To address some of these issues", which ones? Please be more specific or delete.

2/29: what do you mean by "surface controls"?

2/35: Section 1.1? This section numbering is odd. I wouldn't use 1.1 if there is not a 1.2.

3/10: Did you observe penitentes on the glacier? These could be an indication of non-negligible sublimation.

5/24-25: This should be included in the discussion. Maybe total ablation values larger than 11 m w.e. are not realistic.

7/26: Please add the value that you calculated for the slope from Aster.

8/6-8: Please explain better the procedure to calculate k. What value do you obtain? Do you use the water level sensor to calibrate it?

8/13-17: What did you find at AWS2? Is this station more influenced by free atmospheric flow? Is there a predominant wind direction?

8/24: This section should be moved to 3.3, i.e. after presenting results from the EBM.

9/6: "Incoming shortwave radiation …" but you did not measure the incoming longwave radiation.

12/7: Please be consistent in these comparisons. You are not comparing melt rates (which should be given in units of time), but total melt amounts. However, these total melt amounts are not really comparable because they were calculated for different time periods (Juncal Norte Glacier: December-February, Phichillancahue Glacier: January-March, Universidad Glacier: October to March). Is melt at Universidad Glacier really larger than at the other glaciers of the region? Please see main comment 1b.

13/29-30: "The potential for hydropower…" This sounds like a study site description.

13/31: "More studies…" What type of studies? or do you mean more long-term stations?

Figure 8: The description/discussion of this figure is very short.

Figure 11: Consider deleting this figure or extending its description/discussion. The authors state that it shows an efficiently channelized drainage system, but it is not clear from the figure. Maybe you can add a comparison of calibrated k values with those from other glaciers?

**TECHNICAL CORRECTIONS**

Please correct systematically throughout the paper the use of capital letters for glaciers and rivers. Universidad glacier -> Universidad Glacier, Tinguiririca river -> Tinguiririca River. In the revision statement, the authors state that they have done that, but they did not (e.g. in the title).

1/11: melt -> glacier ablation (you also analyze surface sublimation)

1/12: Here and throughout the article: Universidad glacier -> Universidad Glacier

1/12: Here and throughout the article: Tinguiririca river -> Tinguiririca River

1/14: ->distributed temperature-index and runoff routing model (delete melt)

1/14: Please check the grammar of this long sentence or split it in two. "meteorological measurements" are not used to "compare total model modelled glacier melt to river flow measurements". "meteorological measurements" are only used to drive the melt models.

1/18: delete "a contribution"

1/19: total runoff -> streamflow at the outlet of Tinguiririca River Basin

2/2: Please check with the native English speakers (I'm not) in the author's list the use of "which". I think that it is not used correctly in this sentence.

2/13: Same as in 2/2.

2/13: on -> for the

2/13: "current future"?

2/17: "in the glacier ablation zone" -> "on …"

2/30: across -> during?

2/34: Tinguiririca

2/38: November to January -> November and January

3/2: Consider to remove "lower", or is there an "upper tongue"?

3/18-19: Consider to remove the sentence in the parenthesis.

3/26: -> "on the ablation zone"

3/27: -> "on the accumulation zone"

3/34: -> "on the ablation zone"

4/8: "We used the…" for what? Check the grammar.

4/9: using "a regression"?

4/12: What is the resolution of Aster GDEM?

4/14: altitude -> elevation

4/16: We have used -> We used. Please be consistent with the verb tense.

4/31: "calculated ice melt"

4/34: Consider to replace "a" by "M".

5/14: Fig 2 -> Fig 3?

5/15: over an average day -> on an average day?

5/18: Please consider the use of this wording: "While the LR minima are likely to be related to …., the afternoon maxima are potentially caused by the erosion …".

5/22: "we distribute air temperature"

5/28: "Check grammar: "We restrict use of data only up until this date".

5/29: occurs -> occurred.

5/34: "and, as summer precipitation amounts are small, "

6: If I checked correctly, terms P and e_sat were not explained.

6/7: Is k the same as k_0?

6/9: Where -> where

6/9: "Finally,"

6/11: Delete the ",".

6/14: add "where"

6/24: The reference to Hock (2005) is not necessary for that equation.

6/25: Mention that only positive $\Psi$ values are used for that equation.

6/29: Add "where".

7/9: You already described z.

8/2: Consider: "At each grid cell and time step, glacier melt obtained with the ….".

8/10: Consider to reduce the section title. "Meteorological and snow conditions" should be enough.

8/11: Please consider something like: "During the period December-March, air temperature is almost constantly above 0°C at AWS1, but it shows more frequent negative nocturnal values at AWS2."

8/13: "variability, but hourly values".

8/22: Please check the term "high cloud cover", it might be misunderstood as high in elevation.

8/25: "compared to melt"

8/31: Delete: "between 0.29…".

8/35: "in the range of the values estimated by…"

9/19: -> "similar to those at the end of October".

9/25-26: "At the hourly scale, water discharge estimated…"

9/26: Consider: "the values derived from the water pressure sensor".

9/35-36: "between 50% and 66%" (delete "the").

10/3: Delete "Mean total".

10/10: "After the peak in runoff" (delete "the").

10/19: "from glaciers in the central region of Chile".

11/11: "becomes depleted" -> "depletes".

11/14: missing space after Tinguiririca.

11/18: Is this information relevant if the trend was not significant?

11/18-22: Check grammar or split.

11/24: "wheter" -> "whether". Furthermore, if you use "whether", then you need to provide two

alternatives.

11/29: "for e.g."?

11/35-36: Move this to the Introduction or to the Study area.

12/23: estimate -> estimated.

12/34-36: Please split this sentence. It is too long and difficult to follow.

12/34: You didn't really investigate the "climatic conditions". This would require a long time series.

13/7: "greater than melt values reported for other glaciers in central Chile".

13/13: "will be" -> "should be"?

Figure 1:

19/2"Location of Universidad Glacier" (delete "the").

19/2: delete "entire".

19/4: "indicates the locations of stream gauge"?

Figure 2:

"Hourly time series of observed meteorological variables".

Figure 3:

"of hourly lapse rates".

"upper and lower box limits"

Figure 5:

Indicate the date of the ASTER image also in the caption

Figure 8:

c) difference of panels a and b.

Figure 10:

"and the HydroChile…"

Figure 11:

"estimated from the pressure sensor".

Figure 13:

Delete "for reference".

---

## Referee Report (RR3)

Review for Hydrology and Earth System Sciences

**Title: Assessing glacier melt contribution to river runoff at Universidad glacier, central Andes of Chile**

Authors: Bravo, Loriaux, Rivera, Brock

**PAPER SUMMARY AND RECOMMENDATION**

As already reported, I find this article relevant for HESS and interesting, as it provides new information about the glacier contribution to runoff in a region where glaciological studies are relatively scarce.

I think that the authors have properly answered to the last comments of the reviewers regarding the discussion and conclusions. The discussion section has improved with a better description of the uncertainties in the methods. I have two minor points that the authors might want to clarify/include in the text. In my opinion, there are still several problems with the style. I have again provided some suggestions to improve that part.

Finally, I would like to encourage the authors to develop new studies in that region covering longer time periods and/or use more physically-based models able to reproduce glacier ablation and retreat in greater detail.

**MINOR COMMENTS**

You say that AWS2 is located in the accumulation area. Probably you were sure about this at the installation time, but you found that the snow disappeared completely at the site during that summer. I wonder if you should change this for "the upper area". Or maybe mention that that was the accumulation area for the previous years.

9/32: What exactly indicates "an efficiently channelized drainage system flow"? The 6 hours between minimum and maximum? Would you please provide more comments about this? Any reference to support that statement?

**TECHNICAL CORRECTIONS**

1/11: Maybe add "in this region"

1/12: Replace "large" by a number or say "one of the largest in the region"

1/14: I would suggest to move "Total modelled glacier melt…" after the next sentence ("The temperature-index model was calibrated…"

1/17: Maybe replace "is characterized". As you only model one ablation season, it might be an excess to say that the glacier is characterized by the conditions valid then.

1/20: Probably you should mention before that 2009-10 was a dry year with little winter precipitation.

1/29: If you write "in recent years", then I would replace "are increasing" by "have increased"

1/31: Those are independent clauses. Replace "," by ";" or formulate it differently

1/34: add "," after (Mernild et al. 2016)

2/2: I would replace ", which" by "and"

2/6: I would remove "However"

2/9: Please consider to replace "the high basin of the Maipo River" by "the upper Maipo River basin"

2/14: glacier -> glaciers

2/15: Consider to shorten this sentence by starting a new sentence with "Pellicciotti et al. (2008) investigated…"

2/21: improving understanding -> improving the understanding

2/26: "between the humid temperature south and arid north of the country". Please briefly describe the climatic spatial patterns of Chile before in the Introduction. Otherwise this sentence will not be clear for a reader not familiar with the Chilean climate.

2/28: Perhaps be explicit about the two type of models: "using degree-day and energy balance models".

2/37: Cortez -> Cortés. Here and elsewhere.

2/37: "with a runoff peak"

3/1: Add comma before which

3/3: end-of-summer snowline

3/4: Fig. 1c

3/7: I would be more specific and say that proglacial lakes are related to "glacier termini" or "glacier snout" instead of only "glacier"

3/10: Decide for "penitents" or "penitentes"

3/12: Do you have some numbers for the spectacular recession?

3/15: "and identified an increase in surface velocities between 1967 and 1987" An increase respect to what? The analysis starts on 1967.

3/18: The -> This

3/19: Delete: "We focused on…"

3/20: Explain what do "Dirección General de Aguas" and "Dirección Meteorológica de Chile" mean in English for the non-Spanish speakers.

3/9: "using snow density measured at stakes" I guess at stake 1 for the SR50 lowering

3/29: Please mention if the temperature sensor was aspirated.

4/20: Perhaps replace "differentiation" by "recognition" or "identification".

4/21-23: Check the structure of this sentence. It is not very clear. I suggest:

"The MOD10A1 product gives the fractional snow cover for each pixel in the range 1-100. To assure a correct snowline altitude, we assume the presence…. However, …"

4/28: To estimate a FDH for snow, ….

4/37: "in the review article of Hock (2003)"

5/1: "by 24, which resulted in FDH values of 0.29"

5/7: Consider: "we compared melt calculations from a standard degree-day model with those from the DHM"

5/10: at -> with

5/15-17: I am not sure if the structure of this sentence is correct. Please check it.

5/19: Instead of distribute the model, I would say "to extend the model to a distributed scale"

5/21: "considering that melt occurs mostly during the day" Please explain why that is important.

5/25: "during daytime"

5/34: Keep using the same tense: "We restricted…"

6/5: "The turbulent sensible heat…"

7/17: meters -> m

7/18: ",which is …"

8/17: As they describe the same figure, I would merge these 2 paragraphs.

9/1: It is also because the surface albedo is very low

9/6: What do you mean exactly by "small sublimation reflects a melt regime"

9/6: "Snow disappeared…"

9/15: I would say "to estimate glacier melt at the glacier tongue" instead of "total glacier melt" or what do the authors mean exactly?

9/21: As is expected -> As expected?

9/22: on the tongue

9/36-10/2: Please check the grammar/style of these sentences. It sounds a bit strange to me (particularly the use of whereas).

10/9: Maybe change those numbers by "the remaining part"

11/36: "tends to underestimate melt"

12/15: Maybe I was not clear in the previous version. Why reporting a non-significant trend?

12/18: add "likely" before increased.

12/18: In my opinion, the authors are not being precise enough with the terms in these sentences. They say that "the contributing melt area" has increased and then you say "Such increases in glacier melt", but an increase in contributing melt area does not warranty an increase in glacier melt. Most probably yes, but such statements make the article more difficult to read.

12/21: Cortés

12/23: "uncertain" is not the right term here. From your analysis, it is not only uncertain, but also impossible to assess if the "peak water" has been reached.

12/33: in the dry season -> during the dry season

12/35: Please start a new sentence: "On the other hand, to the south of ~37…"

12/36: Replace "which" by "that"

12/36: Not sure if "enhances" is the right verb here. Maybe "allows"?

12/37: Add comma after melting

12/37-38: Local factors, such as …, also contribute…

13/2: context -> reference

13/11: found -> showed

13/34: add comma after Chile

14/2: "are well suited" -> are appropriate for, support the application

15/4: Add volume and issue

16/26: Remove underline in Mernild

23/4: turbulent latent and sensible heat fluxes.

24/3: from Universidad Glacier

25/8: with the distributed degree-hour model

25/8: Delete comma

---

## Author Response (AR2)

**Author's Response**

We appreciate the insightful reviews from the two referees and for the time involved in the detailed revision of our manuscript. Also we appreciate the comments of the editor. We believe this input has greatly improved the quality of our paper. In general, we agree with all comments, and have introduced changes in all sections, and have revised the Discussion where we expanded the discussion of uncertainty as the referees suggested. Furthermore, we conducted a detailed revision of the grammar and typing, and we hope that the revised manuscript meets the criteria for publishing in HESS. In the next sections, we answered point by point the comments of the referees (in bold) and give our responses in normal text, and also marked-up manuscript version showing the changes made.

**Author Response to Referee 1**

**Specific comments**

**1 / 18: does glacier contribution account for 10-13% for the whole period (November 2009 to March 2010) or even 3-34%? do you really differentiate between November to March and December to March? quite confusing! Table 3 does not include November - I guess you have not revised and adapted all sections, do so now!: 1 / 18; 10 / 03; 10 / 9-11; 11 / 06; 13 / 13**

We clarified and simplified this point in all sections. We now use just monthly mean values and removed references to November in the text (1/17-20; 10/18-19; 12/2-3 of the new manuscript).

**3 / 21: the 'mega-drought' conditions in Central Chile are still prevailing, right? considering that we are already in 2017 and you want to emphasize this present effect of drought you could extend "2015-2016" (and, if available, include some latest bibliography) – also in the other sections**

We agree with this comment, and add that the drought extends to 2017 with reference to government agencies in Chile (3 / 21-22). Initially, we stated 2010-2015 as this was the period covered by the most up to date reference on the topic.

**10 / 21-23: the good agreement you mention does not result from data availability itself, improve this Part**

We changed by "This good agreement results from: first, on-glacier measurements of meteorological data at two locations, enabling the use of a local hourly-calibrated lapse rate to extrapolate air temperature inputs to the distributed melt model;" (10/29-31)

**11 / 17-18: is the "positive trend" of 0.3 m³/s/y significant or not? if not, you cannot refer to a "positive trend"**

We clarify this, stating that the trend is not significant (12/14-15)

**11 / 31: start a new sentence "However, the possibility…" and sharpen your argumentation as a) I understood that drought conditions are still going on and b) this sentence calls for a consequence, e. g. "… in the future and therefore more research is needed in order to address these issues."**

We changed this to: "Estimations of the future runoff trend and melt contribution from Universidad Glacier are beyond the scope of this work. However, the possibility of increased persistence and recurrence of droughts in central Chile (Boisier et al., 2016) would increase the hydrological importance of Universidad Glacier in the future and therefore more research is needed in order to address these issues." (12/27-30)

**12 / 07-10: the periods are both three months, from January to March for 2004 and 2005 with corresponding melt values of 4950 mm w.e. (2004) and 3960 mm w.e. (2005)? a little bit confusing**

We apologize for the confusion caused and changed this sentence to: "Brock et al. (2007) estimated cumulative melt of 4950 mm w.e. and 3960 mm w.e in the January to March periods of 2004 and 2005, respectively, at 2000 m asl on Pichillancahue Glacier on Villarrica Volcano further to the south (39°S)."

**12 / 29-30: both studies you are mentioning in the context of the importance of groundwater flows were performed in climatically different regions, the outer tropics in Peru and central Chile. Baraer et al. 2014 refer particularly to the dry season (of each year) - please clarify**

We added addition detail to specify the different regions of these 2 studies, and that Rodriguez et al. (2016) suggest that subsurface storages is the main contributor in winter and fall: "It has been suggested that another source of streamflow during the dry season is groundwater flow i.e. in the outer tropics in Peru, the groundwater contribution to outflow is greater than 24% in all of the analyzed valleys by Baraer et al. (2014). In central Chile, Rodriguez et al. (2016) estimated that contribution associate to subsurface storage in winter and fall is of 60% in Juncal Norte Basin" (13/27-30).

**13 / 32: although it is very likely, please cite your statement "increasing demand for water in the region"**

We changed by:"…considering climate change and the increasing demand for water in the region (Meza et al., 2012)."(14/26)

**Technical corrections**

**1 / 19: better reduce "during November 2009 and March 2010."**
Done.

**2 / 02: Delete (Garreaud, 2013), already cited in line 3**
Done.

**2 / 38: correct "between November and January"**
Done.

**3 / 04: "depending on" the aspect**
Done.

**3 / 18-19: delete "(after March 2010 no more data/observations were obtained from the glacier)", no additional information**
Ok, we deleted this.

**3 / 26-27: three times "installed", avoid redundancies**
New sentence: "Two AWS were installed on the surface of the glacier (Fig. 1). One on the ablation zone (AWS1, 34° 42' S, 70° 20' W, 2650 m asl) and the second one on the accumulation zone (AWS2, 34° 38´ S, 70°19' W, 3626 m asl)."

**4 / 17-18: "dirty" is a very colloquial and "impurities" a very broad term, better write "partially covered by debris and aerosols"**
We changed by term suggested.

**4 / 30: add "up" after "summed"**
Added.

**4 / 31-32: be consistent with the use of units "between 7 mm w.e. °C-1 d-1 and 9 mm w.e. °C-1 d-1" or just "from 7-9 mm w.e. °C-1 d-1"**
We used the first one.

**5 / 16: again "between … and …"**
Corrected.

**5 / 28: use past instead of present "We restricted"**
Done.

**5 / 29: be consistent with other date formats in your manuscript "29 January"**
Ok.

**7 / 14: better write "from a catchment with a total area of 86 km² which is partially covered by…"**
Changed.

**7 / 24: use singular "The geometry... was measured"**
Done.

**8 / 16: again "between … and …"**
Corrected.

**9 / 13-14: avoid redundancies "some melting, with values around…"**
All parts of the glacier experienced melting with values around 1 m w.e. in the upper accumulation area.

**9 / 22: use "16:00 h… 10:00 h" or "4:00 pm … 10:00 am", be consistent with the same format used in 8 / 16-17 and other sections**
Ok, we used the first format.

**10 / 08: correct "Tinguiririca basin" and add "to" after "due"**
Done.

**11 / 14: divide "Tinguiririca basin"**
Done.

**11 / 19: again "between 1975 and 2001"**
Corrected.

**11 / 21-22: better write "positive discharge trends for several rivers in central Chile"**
Done.

**11 / 23: correct "Tinguiririca"**
Done.

**11 / 24: "indicating that"**
Done.

**11 / 24-25: be careful, a typical grammar issue in your manuscript "tends to occur"**
Changed.

**11 / 26: correct "it is uncertain" and write "whether" (avoidable with a normal grammar check!)**
Done.

**11 / 28: correct "is yet to occur"**
Done.

**11 / 30: use plural "Estimations of"**
Ok.

**11 / 37: use the English form for "Peru"**
Ok.

**11 / 39: eliminate "due to increasing latitude" and use plural "reduce"**
Done.

**12 / 01: use plural "Local factors"**
Done.

**13 / 10: add "asl" after "~3800 m"**
Added.

**13 / 28: eliminate first "forestry" and "irrigation", both are redundant**
Done.

**13 / 30: not clear "medium to long term trends" – you refer to future discharge trends, clarify**

Changed by:" will be affected by interannual variability in water supply and future streamflow trends in the medium to long term.

**13 / 31: more precise is "glacier contribution to river discharge"**
Changed.

**Author Response to Referee 2**

**General comments**

*1. Conclusions*
**Some conclusions are confusing and not all of them are properly supported by the obtained results. The authors might consider restructuring the conclusions based on to the proposed objectives. I copy here the conclusions:**

**a. "Good agreement was found between melt estimated from degree-hour and energy balance models, and ablation stake and sonic ranger records at the lower weather station site in the ablation zone, supporting the application of a simple temperature-index method of calculating total glacier melt at this location. The degree-hour model was distributed at the glacier wide scale accounting for hourly variations in the local temperature lapse rate, which tended to be shallower during the daytime, when most melt occurs."**

**This conclusion is very confusing, as it mixes results ("Good agreement was found…") and methods ("The degree-hour model was distributed…"). In my opinion, the most relevant conclusion in this paragraph is that a degree-hour model provides a good simulation of surface lowering at the glacier tongue. This is probably because surface temperature is constantly close to 0°C and negative latent heat fluxes are negligible.**

We understand that this conclusion is confusing. Following the referee's recommendations we changed the text to:

- "The distributed degree-hour model provides a robust simulation of surface melt, especially on the glacier tongue where good agreement was found between melt estimated from the point scale degree-hour model, energy balance model, ablation stake measurements and sonic ranger records. Almost continuously positive air temperatures in the ablation zone between November and March are well suited to the application of a simple temperature index method to calculate glacier melt, however, some melt overestimation was identified for the accumulation zones due to more frequent negative air temperatures at higher elevations." (13/37-38; 14/1-4)

**b. "The ablation regime is dominated by incoming shortwave radiation, with highest melt rates occurring during December to February, and is also characterized by high air temperature which is almost continuously positive on the lower ablation zone between November and March. These climatic conditions result in very high melt totals, which exceed 10 m w. e. melt on the lower tongue and are thus greater than melt values reported for other glaciers in central Chile. This is attributed to the relative insignificance of sublimation to total ablation, and the high insolation due to low cloud cover and**

**latitudinal location, combined with predominantly positive air temperature. Melt totals were much lower in the accumulation area due to lower temperatures and persistent snow cover above about ~3800 m."**

**As incoming longwave radiation was not measured, I'm not sure if the authors can state that incoming shortwave controls the ablation regime. Or what do they mean exactly? That melt correlates to shortwave radiation or that incoming shortwave is the main energy input? Please be more explicit. In fact, at the end of January, incoming shortwave clearly decreases, but the energy available for melt keeps constant (or even increases) (Figure 7). Furthermore, if the regime is dominated by shortwave radiation a good explanation must be provided for the successfully use of a temperature-index model. Would an enhanced-temperature index model (that includes shortwave radiation) improve simulations?**

**The comparison with other glaciers does not support the hypothesis that melt rates on Universidad Glacier are larger than those on other glaciers of the central region. Only one glacier in the central region is used for comparison (Juncal Norte Glacier), but the simulated period was different (I think that Pellicciotti et al. 2014 used data described in Ragettli and Pellicciotti 2012: only the period December 2008 to February 2009). Furthermore, only one season at each glacier is not enough to provide a meaningful comparison. The hypothesis of large melt rates at Universidad Glacier is interesting, but should be better justified. If this is the case, is Universidad Glacier retreating faster than other glaciers? Has this been observed? Is there any other evidence that supports this hypothesis? Surface sublimation on Juncal Norte is also negligible on the glacier tongue (Pellicciotti et al. 2008).**
**"Melt totals were much lower in the accumulation area…" This is obvious. Please delete or explain further.**

We agree that to conclude that the ablation regime is dominated by incoming shortwave radiation is a large assumption considering that we don't have longwave radiation measurements and have deleted the statements to this effect. Also we deleted the assumption that Universidad Glacier has the highest melt values in central Chile, and replaced this with the more cautious statement:

"Hence, Universidad Glacier may be located in a climatic zone which enhances high rates of summer melting as Sagredo and Lowell (2012) suggest in their climate zone classification for Andean glaciers. Local factors such as the large accumulation area and extension of the glacier tongue to a relatively low elevation also contributes to the high melt detected in the lower zone of the glacier (Fig. 8)."(12/36-39)

In the paragraph comparing melt totals at Universidad Glacier with other glaciers in Chile (Juncal Norte and Picchillancahue), we have redone the melt calculations to compare the same months between each study, and made the interpretation more cautious to emphasise the difficulty in comparing different locations in different years:

" Although melt rates cannot be compared directly between different glaciers in different years, two other studies in Chile provide a context for the DDHM results for Universidad Glacier in the 2009-2010 season. Pellicciotti et al. (2014) estimated the total melt in the lower ablation zone of Juncal Norte Glacier (33°S) to be between 5000-6000 mm w.e. in the December 2008 to February 2009 period, which is slightly lower than the total melt of ~5000-6000 mm w.e.

for the equivalent months of 2009-2010 at the AWS1 location on Universidad Glacier. Brock et al. (2007) estimated cumulative melt of 4950 mm w.e. and 3960 mm w.e in the January to March periods of 2004 and 2005, respectively, at 2000 m asl on Pichillancahue Glacier on Villarrica Volcano further to the south (39°S). This location likely represents the maximum ablation on Pichillancahue Glacier due to a continuous thick mantle of insulating tephra covering the glacier below this elevation. The total melt for the equivalent months of 2010 at the AWS1 location (2650 m asl) on Universidad glacier was higher, between 4800-5700 mm w.e., however, comparison of melt at different elevations on these two glaciers should be interpreted with caution."(13/1-9).

Finally we shortened the related point in the conclusion to:

-        "Meteorological conditions result in very high ablation season melt totals, which reach 10 m w. e. on the lower tongue. This finding is attributed to the high insolation due to a low percentage of cloud cover, combined with a predominantly positive air temperature."(14/5-7)

**c. "During the late ablation season, in February and March 15 2010, when other runoff sources such as snowmelt become depleted, the daily contribution of Universidad glacier to total runoff in the Tinguiririca reached as high as 34%."**

**As the authors state that there is a 1 or 2 days of lag between the simulated runoff from Universidad Glacier and streamflow measured at the DGA station (lines 10/15), daily values should not be used to estimate the percentage of glacier contribution. Calculations should be made only at the monthly scale, i.e. the 3rd and 4th column of Table 3 should be deleted.**

We agree and delete all result indicating daily values including the columns of Table 3 and now use only monthly values in the paper (1/17-20; 10/18-19; 12/2-3)

**2. Discussion**

**a. "Forcing temperature-index models with off-glacier data is problematic due to the depression of near-surface air temperature within the glacier boundary layer (Shea and Moore, 2010) under positive ambient temperature conditions"**
**Actually, many researchers argue that on-glacier data is affected by the glacier surface temperature and that temperature-index models should be forced only with off-glacier temperature data, because it is a better indicator of ambient conditions. As the authors did not evaluate the degree-hour model using off-glacier data, they cannot state that on-glacier data works better than off-glacier on this glacier.**

We apologise for being unclear on this point. The reviewer is correct that, at a point scale, temperature-index model melt estimation using off-glacier temperature data has been shown to result in better performance than using on-glacier data. However, our point is that the availability of temperature data at 2 locations of greatly differing altitude enables us to calculate the local on-glacier lapse rate, which is advantageous for a distributed, as opposed to pointscale, melt model. A few recent studies have shown high variability in on-glacier lapse rates and recommended the use of hourly variable lapse rates in distributed melt models, where possible. Another recent study has shown that off-glacier temperature data are a poor indicator of on-glacier lapse rate. This study is available only as a recently awarded PhD thesis and paper currently under review. However, we have included these references in the paper and will take the Editor's advice on whether these references are acceptable for inclusion in the paper.

We have revised the discussion in Section 4.1 to take account of these issues, first:

"This good agreement results from: first, on-glacier measurements of meteorological data at two locations, enabling the use of a local hourly-calibrated lapse rate to extrapolate air temperature inputs to the distributed melt model;" (10/29-31)

And second:

"Forcing distributed temperature-index melt models with off-glacier data can be problematic due to the difficulty in estimating the temperature distribution across the glacier (Shaw, 2017; Shaw et al., submitted). At a point scale, a locally-calibrated temperature-index model forced with off-glacier air temperature data can lead to improvement over use of on-glacier temperature data, due to damping of temperature within the glacier boundary layer (Guðmundsson et al., 2009). However, recent glacier studies have revealed high variability in the local air temperature lapse rate , due to variations in the strength and thickness of the katabatic boundary layer and changes associated with cloud cover and synoptic-scale wind field (Petersen and Pellicciotti, 2011; Petersen et al., 2013; Ayala et al., 2015), which are difficult to account for in off-glacier data. Hence, the availability of temperature measurements for 2 on-glacier locations at different elevations provided suitable data for driving the DDHM." (10/32-37; 11/13)

**b. Others comments**

**The discussion of the uncertainties misses the uncertainty in degree-day factors, which is actually very large. Figure 12 shows that the uncertainty in cumulative runoff at the end of the season is about 7±1 m^3 x 10^7, i.e. about 15%. As the streamflow at the DGA station is very large, this uncertainty is translated as a small percentage of total streamflow (10-13%). Hock (2003) provides a table with a large range of values, how did the authors choose the range between 7 and 9 mm d-1? Did they do some previous tests or used another reference? Another important missing source of uncertainty is the calculation of the snow line from MODIS.**
**The authors should add a discussion about the shortcomings of using a temperature-index model in relation to a more physically-based model. This is one of the aims proposed by the authors (number 2) Do you think that the degree-hour model could be missing something important? Maybe shading effects? Radiation fluxes? Is the model appropriate to simulate long-term mass balance or should only be used to simulate ablation at the glacier tongue?**
**Please include in the discussion the influence of debris-covered areas on the glacier tongue. Are the large observed melt rates substantially reduced by the thermal insulation of debris? Maybe those values above 11 m are not realistic?**

We clarify the estimation of the degree day/hour factor for ice in Section 2.6, including an explanation of an initial calibration using the sonic ranger data:

"We did not have ablation stake measurements in the period when the ice surface was exposed, so we instead calibrated the $F_{DD}$ for ice based on melt estimated from the sonic ranger for the period after the 21 November, when field observations confirmed the site was snow free, to 10 December, the end of the sonic ranger record. The resulting $F_{DD}$ value was close to 8 mm w.e. $°C^{-1} d^{-1}$, but to account for uncertainty due to the short period of ablation data on ice we applied a range of $F_{DD}$ values between 7 mm w.e. $°C^{-1} d^{-1}$ and 9 mm w.e. $°C^{-1} d^{-1}$, which corresponds to the mid-range of values for glacier ice reported in the review of Hock (2003). (4/32-37).

And added text in the discussion (4.1) to emphasise this key source of model uncertainty:

"The key sources of uncertainty in the results are: (a) The degree-hour factor of ice. A lack of stake measurements and only a short period of sonic ranger data on ice means there is some uncertainty in a representative ice $F_{DH}$ value at Universidad Glacier. A range of $F_{DH}$ values between 0.29 mm w.e. $°C^{-1} h^{-1}$ and 0.38 mm w.e. $°C^{-1} h^{-1}$ was applied to account for this uncertainty, but we note that published ice $F_{DD}$ values show a much greater range (Hock, 2003). Figure 7 shows that the accumulated melt of the DHM using an $F_{DH}$ of 0.29 mm w.e. $°C^{-1} h^{-1}$ was similar to the melt estimated by the EBM, stake and sonic range measurements in November. However, at the end of the comparison period (end of January), melt estimated using an $F_{DH}$ of 0.38 mm w.e. $°C^{-1} h^{-1}$ more closely matches the energy balance melt estimation. This translates into an uncertainty of 11% in the cumulative runoff from the DDHM at the end of the ablation season (Fig. 11). As the streamflow at the DGA station is large, this ice melt uncertainty contributes only a small percentage of total streamflow (3%)."(11/8-16)

We have added some discussion about the uncertainty in the snow line altitude estimation using MODIS data and annotated Fig. 7 with the lag in the snow/ice transition date between MODIS and field observations at the location of AWS1.

"(b) The snow line altitude derived from the MOD10A1 product. Although glacier surface characteristics on the tongue allow differentiation between ice and snow, the resolution of the snow product is similar to the width of the glacier tongue. A lag of 10 to 12 days was found between the MOD10A1 product and field observations of the transition from snow to ice at the AWS1 site (Fig. 7). Furthermore, in the highest zone of the glacier, fewer debris and aerosols cover the ice surface, making it harder to distinguish between ice and snow, which could have led to errors in identifying surface type."(11/16-21)

We added more information about the general limitation of this kind of model and some specific issues as the referee suggests, related to debris-cover (section 4.1):

"Although we consider the model outputs to be robust, it is important to bear in mind that empirical temperature-index models do not attempt to simulate the real physical processes of glacier ablation, and the DDHM ignores other influences on rates and spatial patterns of ablation, such as topographic shading, blowing snow, debris-cover and subsurface fluxes. Hence, the DDHM may not be suitable for longer term mass balance studies where climatic and surface factors may undergo change."(11/4-7)

And:

"c) DDHM melt estimates were not adjusted for the effects of moraine and patchy distributed debris in the ablation zone (Fig. 1). The moraines are of substantial thickness on lower areas of the tongue and likely to reduce ablation below the highest values shown in Fig. 8 in the terminus zone. However, other areas of the ablation zone are affected by a thin and patchy layer of debris or aerosol, which is likely to increase ablation through local albedo reduction (Fyffe et al., 2014). Although, quantification of the effects of debris on melt is beyond the scope of this study it would be expected that impacts of thick morainic debris and thin patchy debris elsewhere will tend to compensate in overall melt estimations for the glacier."(11/21-26)

**MINOR COMMENTS**
**The Introduction could benefit from a better structure in which climate, hydrology and economic characteristics are separately described.**
**Please check the use of terms "runoff" and "streamflow". I think that they are not the same: Runoff is the portion of precipitation or melt that does not infiltrate or evaporate. Streamflow is the runoff of surface water through a channel.**

We checked the terms and made changes according referee suggestions, including the paper title.

**2/22: Why is 40°S important?**

We deleted this.

**2/27: "To address some of these issues", which ones? Please be more specific or delete.**

Deleted.

**2/29: what do you mean by "surface controls"?**

We changed by "to identify the principal meteorological drivers of ablation and their patterns and trends during a full ablation season"

**2/35: Section 1.1? This section numbering is odd. I wouldn't use 1.1 if there is not a 1.2.**

Ok, we moved this section to methods.

**3/10: Did you observe penitentes on the glacier? These could be an indication of non-negligible sublimation.**

We don't observed penitents in the glacier. We added observations of Lliboutry (1958) to the Study Area section.

**5/24-25: This should be included in the discussion. Maybe total ablation values larger than 11 m w.e. are not realistic.**

We agree and added to discussion.

**7/26: Please add the value that you calculated for the slope from Aster.**

Added.

**8/6-8: Please explain better the procedure to calculate k. What value do you obtain? Do you use the water level sensor to calibrate it?**

We explain better this issue. We use the water level sensor to calibrate it.

**8/13-17: What did you find at AWS2? Is this station more influenced by free atmospheric flow? Is there a predominant wind direction?**

We added the wind direction of AWS2 in the description but not as a new Figure.

**8/24: This section should be moved to 3.3, i.e. after presenting results from the EBM.**

Ok, section moved.

**9/6: "Incoming shortwave radiation …" but you did not measure the incoming longwave radiation.**

We agree, please see response to General Comments.

**12/7: Please be consistent in these comparisons. You are not comparing melt rates (which should be given in units of time), but total melt amounts. However, these total melt amounts are not really comparable because they were calculated for different time periods (Juncal Norte Glacier: December-February, Phichillancahue Glacier: January-March, Universidad Glacier: October to March). Is melt at Universidad Glacier really larger than at the other glaciers of the region? Please see main comment 1b.**

We agree, but we maintain the comparison, but now for the same periods (months). Please see our response to the General Comments. This is due to give a context of our results using previous studies. However we agree that the conclusion of Universidad Glacier has the larger melt in the region does not find support in our results.

**13/29-30: "The potential for hydropower…" This sounds like a study site description.**

We changed this to "Hydropower generation on the Tinguiririca River at La Higuera and La Confluencia (Pelto, 2010), will be affected by interannual variability…".

**13/31: "More studies…" What type of studies? or do you mean more long-term stations?**

We changed this to: "Finally, more long-term high elevation stations in the Andes are necessary to establish the inter-annual variability of glacier contribution to river discharge in order to help manage future water availability, considering climate change and the increasing demand for water in the region (Meza et al., 2012)."

**Figure 8: The description/discussion of this figure is very short.**

We don't think it is necessary to add more text at this point as the key points are displayed visually by the figure. However, Fig. 8 does inform the later discussion at various points, e.g. in assessing the impact of debris areas on melt rates.

**Figure 11: Consider deleting this figure or extending its description/discussion. The authors state that it shows an efficiently channelized drainage system, but it is not clear from the figure. Maybe you can add a comparison of calibrated k values with those from other glaciers?**

We deleted this Figures as referee suggests.

**TECHNICAL CORRECTIONS**
**Please correct systematically throughout the paper the use of capital letters for glaciers and rivers. Universidad glacier -> Universidad Glacier, Tinguiririca river -> Tinguiririca River. In the revision statement, the authors state that they have done that, but they did not (e.g. in the title).**
Done

**1/11: melt -> glacier ablation (you also analyze surface sublimation)**
Done.

**1/12: Here and throughout the article: Universidad glacier -> Universidad Glacier**
Done.

**1/12: Here and throughout the article: Tinguiririca river -> Tinguiririca River**
Done.

**1/14: ->distributed temperature-index and runoff routing model (delete melt)**
Done.

**1/14: Please check the grammar of this long sentence or split it in two. "meteorological measurements" are not used to "compare total model modelled glacier melt to river flow measurements". "meteorological measurements" are only used to drive the melt models.**
Changed to:
"We used meteorological measurements from two automatic weather stations installed on the glacier to drive a distributed temperature-index and runoff routing model. Total modelled glacier melt is compared with river flow measurements at three sites located between 0.5 and 50 km downstream."

**1/18: delete "a contribution"**
Deleted.

**1/19: total runoff -> streamflow at the outlet of Tinguiririca River Basin**
Changed.

**2/2: Please check with the native English speakers (I'm not) in the author's list the use of "which". I think that it is not used correctly in this sentence.**
Checked.

**2/13: Same as in 2/2.**
Checked.

**2/13: on -> for the**

Done.

**2/13: "current future"?**
Changed by "future glacier contribution".

**2/17: "in the glacier ablation zone" -> "on …"**
Done.

**2/30: across -> during?**
Done.

**2/34: Tinguiririca**
Done

**2/38: November to January -> November and January**
Done.

**3/2: Consider to remove "lower", or is there an "upper tongue"?**
Removed.

**3/18-19: Consider to remove the sentence in the parenthesis.**
Removed.

**3/26: -> "on the ablation zone"**
Done.

**3/27: -> "on the accumulation zone"**
Done.

**3/34: -> "on the ablation zone"**
Done.

**4/8: "We used the…" for what? Check the grammar.**
Corrected.

**4/9: using "a regression"?**

Done.

**4/12: What is the resolution of Aster GDEM?**
30 m added.

**4/14: altitude -> elevation**
Changed.

**4/16: We have used -> We used. Please be consistent with the verb tense.**
Ok.

**4/31: "calculated ice melt"**

Done.

**4/34: Consider to replace "a" by "M".**
Replaced as suggested

**5/14: Fig 2 -> Fig 3?**
Changed.

**5/15: over an average day -> on an average day?**
Changed.

**5/18: Please consider the use of this wording: "While the LR minima are likely to be related to …., the afternoon maxima are potentially caused by the erosion …".**
Changed:
"While the LR minima are likely to be related to the strengthening of katabatic flow during the daytime (Petersen and Pellicciotti, 2011), the afternoon maximum is potentially caused by the erosion of the katabatic boundary layer on the lower glacier tongue, due to warm air advection from bare rock surfaces at the glacier sides and proglacial area (van de Broeke, 1997; Ayala et al., 2015)."

**5/22: "we distribute air temperature"**
Done.

**5/28: "Check grammar: "We restrict use of data only up until this date".**
Changed.

**5/29: occurs -> occurred.**
Changed.

**5/34: "and, as summer precipitation amounts are small, "**
Changed.

**6: If I checked correctly, terms P and e_sat were not explained.**
Added.

**6/7: Is k the same as k_0?**
Yes, changed.

**6/9: Where -> where**
Done.

**6/9: "Finally,"**
Changed.

**6/11: Delete the ",".**
Done.

**6/14: add "where"**
Added.

**6/24: The reference to Hock (2005) is not necessary for that equation.**
Deleted

**6/25: Mention that only positive Ψ values are used for that equation.**
Added.

**6/29: Add "where".**
Added.

**7/9: You already described z.**
Ok, deleted.

**8/2: Consider: "At each grid cell and time step, glacier melt obtained with the ....".**
Changed as referee suggested.

**8/10: Consider to reduce the section title. "Meteorological and snow conditions" should be enough.**
Done.

**8/11: Please consider something like: "During the period December-March, air temperature is almost constantly above 0°C at AWS1, but it shows more frequent negative nocturnal values at AWS2."**
Changes as referee suggested.

**8/13: "variability, but hourly values".**
Changed.

**8/22: Please check the term "high cloud cover", it might be misunderstood as high in elevation.**
Changed by: "a high percentage of cloud cover (greater than 30%) affected snowline detection"

**8/25: "compared to melt"**
Changed.

**8/31: Delete: "between 0.29...".**
"…fall within the DHM range for $F_{DH}$ values between 0.29 mm w.e. h$^{-1}$ °C$^{-1}$ and 0.38 mm w.e. h$^{-1}$ °C$^{-1}$.".

**8/35: "in the range of the values estimated by…"**
Changed.

**9/19: -> "similar to those at the end of October".**
Changed.

**9/25-26: "At the hourly scale, water discharge estimated…"**
Changed.

**9/26: Consider: "the values derived from the water pressure sensor".**
Changed as referee suggests.

**9/35-36: "between 50% and 66%" (delete "the").**
Deleted.

**10/3: Delete "Mean total".**
Deleted.

**10/10: "After the peak in runoff" (delete "the").**
Deleted

**10/19: "from glaciers in the central region of Chile".**
Changed.

**11/11: "becomes depleted" -> "depletes".**
Changed.

**11/14: missing space after Tinguiririca.**
Corrected.

**11/18: Is this information relevant if the trend was not significant?**
Probably not but we maintain as previous studies.

**11/18-22: Check grammar or split.**
Corrected.

**11/24: "wheter" -> "whether". Furthermore, if you use "whether", then you need to provide two alternatives**.
Deleted "whether".

**11/29: "for e.g."?**
Corrected.

**11/35-36: Move this to the Introduction or to the Study area.**
Corrected.

**12/23: estimate -> estimated.**
Changed.

**12/34-36: Please split this sentence. It is too long and difficult to follow.**
Changed to: "In this study, we have investigated the meteorological conditions, ablation and melt water contribution to downstream river flow of Universidad Glacier, located in central Chile during the 2009-2010 summer ablation season. We used a point scale energy balance and a distributed degree-hour melt model, driven by data from two on-glacier weather stations. The main outcomes of this work are:".

**12/34: You didn't really investigate the "climatic conditions". This would require a long time series.**
Agree, corrected.

**13/7: "greater than melt values reported for other glaciers in central Chile".**
Deleted.

**13/13: "will be" -> "should be"?**
Corrected.

**Figure 1:**
**19/2"Location of Universidad Glacier" (delete "the").**
**19/2: delete "entire".**
**19/4: "indicates the locations of stream gauge"?**
Done.

**Figure 2:**
**"Hourly time series of observed meteorological variables".**
Done.

**Figure 3:**
**"of hourly lapse rates".**
**"upper and lower box limits"**
Done.

**Figure 5:**
**Indicate the date of the ASTER image also in the caption**
Added.

**Figure 8:**
**c) difference of panels a and b.**
Corrected.

**Figure 10:**
**"and the HydroChile…"**
Corrected.

**Figure 11:**
**"estimated from the pressure sensor".**
Figure deleted.

**Figure 13:**
**Delete "for reference".**
Deleted (new Figure 12).

[revised manuscript text omitted]

---

## Author Response (AR3)

**AUTHOR`S RESPONSE**

We appreciate the insightful last review of referee 2 and for the time involved in the detailed revision of our manuscript. We also appreciate the words of the referee encourage us to make more research in the Andes. We keep in mind that longer period of studies and more sophisticated physically-based models are necessary to obtain a complete understanding of the dynamics of ablation along the Andean glaciers. Also we want to thanks to the Editor for his recommendations and the time involved in the review of this work. In the next sections, we answered point by point the comments of the referee 2 (in bold) and give our responses in normal text. Also is attached the marked up version of the manuscript with the new corrections.

**MINOR COMMENTS**

**You say that AWS2 is located in the accumulation area. Probably you were sure about this at the installation time, but you found that the snow disappeared completely at the site during that summer. I wonder if you should change this for "the upper area". Or maybe mention that that was the accumulation area for the previous years.**
We agree with this; we specify that AWS2 is close to ELA in the upper part to avoid confusing, not necessary at the accumulation zone: "Two AWS were installed on the surface of the glacier (Fig. 1). One on the ablation zone (AWS1, 34° 42' S, 70° 20' W, 2650 m asl) and the second one on the upper zone close to the ELA (AWS2, 34° 38´ S, 70°19' W, 3626 m asl).

**9/32: What exactly indicates "an efficiently channelized drainage system flow"? The 6 hours between minimum and maximum? Would you please provide more comments about this? Any reference to support that statement?**
We mean that subglacial channelized is efficiently. High amplitude with maximum after solar noon are indication of glacier ice melt which is routed through large subglacial channels that have enlarged in response to melting (Willis, 2011). To clarify we changed to:
" Discharge peaked typically at 16:00 h, from a minimum at 10:00 h which, considering the large size of the glacier, indicates an efficiently subglacial channelized drainage system flow typically of periods of dominant glacier ice melting (Willis, 2011)."

**TECHNICAL CORRECTIONS**

**1/11: Maybe add "in this region"**
Added

**1/12: Replace "large" by a number or say "one of the largest in the region"**
Replaced

**1/14: I would suggest to move "Total modelled glacier melt…" after the next sentence ("The temperature-index model was calibrated…"**
Moved

**1/17: Maybe replace "is characterized". As you only model one ablation season, it might be an excess to say that the glacier is characterized by the conditions valid then.**
We agree and changed by "shows"

**1/20: Probably you should mention before that 2009-10 was a dry year with little winter precipitation.**
Ok

**1/29: If you write "in recent years", then I would replace "are increasing" by "have increased"**
Changed

**1/31: Those are independent clauses. Replace "," by ";" or formulate it differently**

Replaced

**1/34: add "," after (Mernild et al. 2016)**
Added

**2/2: I would replace ", which" by "and"**
Replaced

**2/6: I would remove "However"**
Removed

**2/9: Please consider to replace "the high basin of the Maipo River" by "the upper Maipo River basin"**
Replaced

**2/14: glacier -> glaciers**

Replaced

**2/15: Consider to shorten this sentence by starting a new sentence with "Pellicciotti et al. (2008) investigated…"**
Changed as you suggested

**2/21: improving understanding -> improving the understanding**
Changed

**2/26: "between the humid temperature south and arid north of the country". Please briefly describe the climatic spatial patterns of Chile before in the Introduction. Otherwise this sentence will not be clear for a reader not familiar with the Chilean climate.**
We think that the spatial pattern is implicit in this sentence. Otherwise a very general description of the climate of Chile it can turn very disconnected of the topics discussed in the Introduction section.

**2/28: Perhaps be explicit about the two type of models: "using degree-day and energy balance models".**
Agree. We explicit both models as you suggests.

**2/37: Cortez -> Cortés. Here and elsewhere**.
Replaced

**2/37: "with a runoff peak"**
Added

**3/1: Add comma before which**
Added

**3/3: end-of-summer snowline**
Changed

**3/4: Fig. 1c**
Added

**3/7: I would be more specific and say that proglacial lakes are related to "glacier termini" or "glacier snout" instead of only "glacier"**
Ok, added "glacier termini"

**3/10: Decide for "penitents" or "penitentes"**
"penitents"

**3/12: Do you have some numbers for the spectacular recession?**
1 km

**3/15: "and identified an increase in surface velocities between 1967 and 1987" An increase respect to what? The analysis starts on 1967.**
This mean that velocities in 1985, 1986 and 1987 are higher than 1967, as the authors (Wilson et al, 2016) mentioned in the abstract and conclusion and Fig. 6 of their work.

**3/18: The -> This**
Changed

**3/19: Delete: "We focused on…"**
Phrase deleted

**3/20: Explain what do "Dirección General de Aguas" and "Dirección Meteorológica de Chile" mean in English for the non-Spanish speakers.**
Added: "…2017 according to data from the Dirección General de Aguas de Chile (Chilean Directorate of Water Resources, DGA) and Dirección Meteorológica de Chile (National Weather Service, DMC).

**3/9: "using snow density measured at stakes" I guess at stake 1 for the SR50 lowering**
At the site of each stake. The lowering detected in the stake is converted to w.e. using density measured at the same site and in the same period.

**3/29: Please mention if the temperature sensor was aspirated.**
We mentioned that is not aspirated

**4/20: Perhaps replace "differentiation" by "recognition" or "identification".**
Replace by "identification"

**4/21-23: Check the structure of this sentence. It is not very clear. I suggest:**
**"The MOD10A1 product gives the fractional snow cover for each pixel in the range 1-100. To assure a correct snowline altitude, we assume the presence…. However, …"**
Ok, changed as you suggests

**4/28: To estimate a FDH for snow, ….**
Changed

**4/37: "in the review article of Hock (2003)"**
Changed

**5/1: "by 24, which resulted in FDH values of 0.29"**
Changed

**5/7: Consider: "we compared melt calculations from a standard degree-day model with those from the DHM"**
Ok, we changed

**5/10: at -> with**
Ok

**5/15-17: I am not sure if the structure of this sentence is correct. Please check it.**
Checked

**5/19: Instead of distribute the model, I would say "to extend the model to a distributed scale"**
Agree, we changed as you suggests

**5/21: "considering that melt occurs mostly during the day" Please explain why that is important.**
We agree that this sentence is confusing in this context. We decide delete this sentence.

**5/25: "during daytime"**
Changed

**5/34: Keep using the same tense: "We restricted…"**

Changed

**6/5: "The turbulent sensible heat…"**

Added "turbulent"

**7/17: meters -> m**
Changed

**7/18: ",which is …"**
Comma added

**8/17: As they describe the same figure, I would merge these 2 paragraphs.**
Agree

**9/1: It is also because the surface albedo is very low**
Added

**9/6: What do you mean exactly by "small sublimation reflects a melt regime"**
That ablation processes is dominated by melt. We changed by:
"Sublimation represents a small percentage (2.8%) of the total ablation calculated with the EBM reflecting the predominantly positive air temperatures and, hence, that ablation is dominated by melt."

**9/6: "Snow disappeared…"**
Changed

**9/15: I would say "to estimate glacier melt at the glacier tongue" instead of "total glacier melt" or what do the authors mean exactly?**
Agree. Model and measurement comparison support the model in the tongue

**9/21: As is expected -> As expected?**
Changed

**9/22: on the tongue**
Changed

**9/36-10/2: Please check the grammar/style of these sentences. It sounds a bit strange to me (particularly the use of whereas).**
To avoid confusion we changed by "A sudden jump in HydroChile and pressure sensor values occurred around this date, most likely due to this earthquake. The pressure sensor derived values were adjusted for the change in water height, however we rejected data from the HydroChile station after the earthquake."

**10/9: Maybe change those numbers by "the remaining part"**
Changed

**11/36: "tends to underestimate melt"**
Changed

**12/15: Maybe I was not clear in the previous version. Why reporting a non-significant trend?**
Ok, we finally decide delete this paragraph to avoid confusions

**12/18: add "likely" before increased.**
Added

**12/18: In my opinion, the authors are not being precise enough with the terms in these sentences. They say that "the contributing melt area" has increased and then you say "Such increases in glacier melt", but an increase in contributing melt area does not warranty an increase in glacier melt. Most probably yes, but such statements make the article more difficult to read.**

As the referee note we use the word "might" to identify a possible cause-effect. However we added "area" to clarify: "Such increases in glacier melt area might explain the positive discharge trends for several rivers in central Chile as suggested by Casassa et al. (2009)."

**12/21: Cortés**
Changed

**12/23: "uncertain" is not the right term here. From your analysis, it is not only uncertain, but also impossible to assess if the "peak water" has been reached.**
Agree, changed by "…it is impossible to assess…"

**12/33: in the dry season -> during the dry season**
Changed

**12/35: Please start a new sentence: "On the other hand, to the south of ~37…"**
Ok, changed as you suggests

**12/36: Replace "which" by "that"**
Changed

**12/36: Not sure if "enhances" is the right verb here. Maybe "allows"?**
Agree, changed by "allows"

**12/37: Add comma after melting**
Added

**12/37-38: Local factors, such as …, also contribute…**
Commas added

**13/2: context -> reference**
Changed

**13/11: found -> showed**
Changed

**13/34: add comma after Chile**
Added

**14/2: "are well suited" -> are appropriate for, support the application**
Ok, changed by "appropriate for…"

**15/4: Add volume and issue**
Added

**16/26: Remove underline in Mernild**
Removed

**23/4: turbulent latent and sensible heat fluxes.**
Added "turbulent"

**24/3: from Universidad Glacier**
Changed

**25/8: with the distributed degree-hour model**
Added

**25/8: Delete comma**

Deleted

[revised manuscript text omitted]